# CCL28 modulates neutrophil responses during infection with mucosal pathogens

Gregory T Walker[1†], Araceli Perez-Lopez[1,2,3†], Steven Silva[1], Michael H Lee[1], Elisabet Bjånes[1], Nicholas Dillon[1,4], Stephanie L Brandt[1], Romana R Gerner[1,5], Karine Melchior[1,6], Grant J Norton[1], Felix A Argueta[1], Frenchesca Dela Pena[1], Lauren Park[1], Victor A Sosa-Hernandez[7,8], Rodrigo Cervantes-Diaz[7,8], Sandra Romero-Ramirez[7,9], Monica Cartelle Gestal[10], Jose L Maravillas-Montero[7], Sean-Paul Nuccio[1,2], Victor Nizet[1,11,12], Manuela Raffatellu[1,2,12,13*]

[1]Division of Host-Microbe Systems & Therapeutics, Department of Pediatrics, University of California, San Diego, La Jolla, United States; [2]Department of Microbiology and Molecular Genetics, University of California Irvine, Irvine, United States; [3]Biomedicine Research Unit, Facultad de Estudios Superiores Iztacala, Universidad Nacional Autónoma de México, Tlalnepantla, Mexico; [4]Department of Biological Sciences, University of Texas at Dallas, Richardson, United States; [5]School of Life Sciences, ZIEL - Institute for Food and Health, Freising-Weihenstephan, Technical University of Munich, Munich, Germany; [6]School of Pharmaceutical Sciences, São Paulo State University (UNESP), Araraquara, São Paulo, Brazil; [7]Red de Apoyo a la Investigación, Universidad Nacional Autónoma de México and Instituto Nacional de Ciencias Médicas y Nutrición Salvador Zubirán, México City, Mexico; [8]Departamento de Biomedicina Molecular, Centro de Investigación y de Estudios Avanzados del Instituto Politécnico Nacional, Mexico City, Mexico; [9]Facultad de Medicina, Universidad Nacional Autónoma de México, Mexico City, Mexico; [10]Department of Microbiology and Immunology, Louisiana State University Health Sciences Center at Shreveport, Shreveport, United States; [11]Skaggs School of Pharmacy and Pharmaceutical Sciences, University of California San Diego, La Jolla, United States; [12]Center for Microbiome Innovation, University of California San Diego, La Jolla, United States; [13]Chiba University-UC San Diego Center for Mucosal Immunology, Allergy, and Vaccines (CU-UCSDcMAV), La Jolla, United States

*For correspondence:
manuelar@ucsd.edu

†These authors contributed equally to this work

Competing interest: The authors declare that no competing interests exist.

**Abstract** The chemokine CCL28 is highly expressed in mucosal tissues, but its role during infection is not well understood. Here, we show that CCL28 promotes neutrophil accumulation in the gut of mice infected with *Salmonella* and in the lung of mice infected with *Acinetobacter*. Neutrophils isolated from the infected mucosa expressed the CCL28 receptors CCR3 and, to a lesser extent, CCR10, on their surface. The functional consequences of CCL28 deficiency varied between the two infections: *Ccl28−/−* mice were highly susceptible to *Salmonella* gut infection but highly resistant to otherwise lethal *Acinetobacter* lung infection. In vitro, unstimulated neutrophils harbored pre-formed intracellular CCR3 that was rapidly mobilized to the cell surface following phagocytosis or inflammatory stimuli. Moreover, CCL28 stimulation enhanced neutrophil antimicrobial activity, production of reactive oxygen species, and formation of extracellular traps, all processes largely dependent on CCR3. Consistent with the different outcomes in the two infection models, neutrophil stimulation with CCL28 boosted the killing of *Salmonella* but not *Acinetobacter*. CCL28 thus plays a critical role in the immune response to mucosal pathogens by increasing neutrophil accumulation and activation, which can enhance pathogen clearance but also exacerbate disease depending on the mucosal site and the infectious agent.

## Editor's evaluation

This important study provides compelling evidence that CCL28 plays a crucial role in regulating neutrophil function and host defense during mucosal infections, with CCL28 deficiency leading to greater susceptibility to *Salmonella* gut infections and enhanced resistance to Acinetobacter lung infections. The data convincingly shows that CCL28, through CCR3, regulates neutrophil functions such as reactive oxygen species production and extracellular trap formation, influencing pathogen clearance and highlighting its context-dependent impact on immunity.

## Introduction

Chemokines comprise a family of small chemoattractant proteins that play important roles in diverse host processes including chemotaxis, immune cell development, and leukocyte activation (*Zlotnik and Yoshie, 2000*; *Zlotnik et al., 2011*; *Charo and Ransohoff, 2006*). The chemokine superfamily includes 48 human ligands and 19 receptors, classified into subfamilies (CC, CXC, C, and $CX_3C$) depending on the location of the cysteines in their sequence (*Nomiyama et al., 2013*; *Hughes and Nibbs, 2018*). Four chemokines predominate in mucosal tissues: CCL25, CCL28, CXCL14, and CXCL17 (*Hernández-Ruiz and Zlotnik, 2017*).

CCL28, also known as Mucosae-associated Epithelial Chemokine, belongs to the CC (or β-chemokine) subclass, and is constitutively produced in mucosal tissues including the digestive system, respiratory tract, and female reproductive system (*Mohan et al., 2017*). Although best studied for its homeostatic functions, CCL28 can also be induced under inflammatory conditions and is thus considered a dual function chemokine (*Mohan et al., 2017*).

CCL28 signals via two receptors: CCR3 and CCR10 (*Pan et al., 2000*). During homeostasis in mice, CCL28 provides a chemotactic gradient for $CCR10^+$ B and T cells and guides the migration of $CCR10^+$ IgA plasmablasts to the mammary gland and other tissues (*Mohan et al., 2017*; *Burkhardt et al., 2019*; *Matsuo et al., 2018*). In a disease context, CCL28 has been best studied in allergic airway inflammation. High CCL28 levels are present in airway biopsies from asthma patients (*O'Gorman et al., 2005*), and $CCR3^+$ and $CCR10^+$ cells are recruited to the airways in a CCL28-dependent fashion in murine asthma models (*John et al., 2005*; *English et al., 2006*).

In the human gut, CCL28 is upregulated during inflammation of the gastric mucosa in *Helicobacter pylori*-infected patients (*Hansson et al., 2008*) and in the colon of patients with ulcerative colitis, a prominent form of inflammatory bowel disease (*Lee et al., 2021*; *Ogawa et al., 2004*). In the mouse gut, CCL28 production is increased in the dextran sulfate sodium model of colitis (*Matsuo et al., 2018*). Epithelial cells are an important source of CCL28 (*Lee et al., 2021*; *Ogawa et al., 2004*), and its expression can be induced by stimulation of cultured airway or intestinal epithelial cells with the proinflammatory cytokines interleukin (IL)-1α, IL-1β, or tumor necrosis factor (TNF)-α, or following *Salmonella* infection of cultured HCA-7 colon carcinoma cells (*Ogawa et al., 2004*).

Collectively, a variety of studies have postulated that CCL28 is an important chemokine in inflammatory diseases, ranging from asthma to ulcerative colitis, and during the immune response to infection. Yet, CCL28 function remains understudied, largely because $Ccl28^{-/-}$ mice have only recently been described (*Burkhardt et al., 2019*; *Matsuo et al., 2018*). Here, we investigate the function and underlying mechanism of CCL28 during the mucosal response to infection.

By comparing infection in $Ccl28^{-/-}$ mice and their wild-type littermates, we discovered a key role for CCL28 in promoting neutrophil accumulation to the gut during infection with *Salmonella enterica* serovar Typhimurium (STm) and to the lung during infection with multidrug-resistant *Acinetobacter baumannii* (Ab). Neutrophils isolated from the infected mucosal sites harbored CCL28 receptors, particularly CCR3, on their surface. In vitro, CCR3 was stored intracellularly, and was rapidly detectable on the neutrophil surface upon stimulation with proinflammatory molecules or in response to phagocytosis. Neutrophil stimulation of CCL28 resulted in enhanced neutrophil antimicrobial activity against STm, increased production of reactive oxygen species (ROS), and enhanced formation of neutrophil extracellular traps (NETs), all processes that help control infection but also cause extensive tissue damage. We conclude that CCL28 plays a previously unappreciated role in the innate immune response to mucosal pathogens by regulating neutrophil accumulation and activation.

## Results

### CCL28-mediated responses limit *Salmonella* gut colonization and systemic dissemination

We investigated CCL28 activity during gastrointestinal infection with STm by using the well-established streptomycin-treated C57BL/6 mouse model of colitis (*Barthel et al., 2003*; *Walker et al., 2023*). At day 4 post-infection (4 dpi) with STm, we observed a ~fourfold increase of CCL28 by enzyme-linked immunosorbent assay (ELISA) analysis of feces from wild-type mice relative to uninfected controls (*Figure 1A*). In a prior preliminary study, we found that *Ccl28*$^{-/-}$ mice infected with STm exhibited increased lethality compared to their wild-type littermates beginning at day 1 post-infection (*Burkhardt et al., 2019*). To further elucidate the impact of CCL28 on STm infection dynamics and host responses earlier in the course of infection (2–3 dpi), we examined STm colony-forming units (CFU) in the gastrointestinal contents and extraintestinal tissues. Although there was no significant difference in gastrointestinal CFU between wild-type and *Ccl28*$^{-/-}$ mice (*Figure 1B* and *Figure 1—figure supplement 1A*), higher CFU were observed in extraintestinal tissues by 2 dpi (*Figure 1—figure supplement 1B*). By 3 dpi, significantly higher CFU were recovered from the Peyer's patches, the mesenteric lymph nodes, and systemic sites (bone marrow and spleen) of *Ccl28*$^{-/-}$ mice (*Figure 1C*), indicating that the CCL28 is essential for limiting extraintestinal STm dissemination. In contrast, when bypassing the gut and infecting mice intraperitoneally with STm, we also observed a ~fourfold increase in serum CCL28 (*Figure 1—figure supplement 2A*), but equal numbers of STm CFU were recovered from the spleen, liver, and blood of both wild-type and *Ccl28*$^{-/-}$ mice at 4 dpi (*Figure 1—figure supplement 2B*). These results suggest that CCL28 helps control STm infection at its origin in the gut mucosa, reducing dissemination to other sites.

### CCL28 promotes neutrophil accumulation to the gut during *Salmonella* infection

CCL28 has direct antimicrobial activity against some bacteria (e.g., *Streptococcus mutans* and *Pseudomonas aeruginosa*) and fungi (e.g., *Candida albicans*) (*Hieshima et al., 2003*), but concentrations up to 1 µM did not substantially inhibit wild-type STm. However, CCL28 produced multilog-fold CFU reductions in *Escherichia coli* K12 or a STm Δ*phoQ* mutant known to be more susceptible to antimicrobial peptide killing (*Groisman, 2001*; *Figure 1—figure supplement 2C*). Therefore, the direct antimicrobial activity of CCL28 does not explain the lower STm colonization in wild-type mice compared to *Ccl28*$^{-/-}$ mice.

During homeostasis, CCL28 exhibits chemotactic activity in the gut mucosa toward CD4$^+$ and CD8$^+$ T cells and IgA-producing B cells (*Mohan et al., 2017*; *Burkhardt et al., 2019*; *Matsuo et al., 2018*). However, immune cell profiling in the intestines (using the flow cytometry gating strategy presented in *Figure 1—figure supplement 3*) revealed similar B cell and CD4$^+$ and CD8$^+$ T cell numbers in both wild-type and *Ccl28*$^{-/-}$ mice during homeostasis and STm infection (*Figure 1—figure supplement 4A–C*). Neutrophils are crucial in the host response to STm (reviewed in *Perez-Lopez et al., 2016*), and neutropenia increases infection severity in both mice and humans (*Bhatti et al., 1998*; *Yaman et al., 2018*; *Vassiloyanakopoulos et al., 1998*; *Fierer, 2001*). Strikingly, we observed increased neutrophil abundance in the intestinal tissues of wild-type mice during colitis, but ~50% fewer neutrophils (CD11b$^+$ Ly6G$^+$ cells) were isolated from the gut of *Ccl28*$^{-/-}$ mice 2 and 3 days after STm infection (*Figure 1D, E*). Concurrent neutrophil counts in the blood and bone marrow were similar between infected *Ccl28*$^{-/-}$ mice and wild-type mice (*Figure 1—figure supplement 5A*), indicating a defect in the accumulation of neutrophils at the mucosal site of infection and excluding a defect in granulopoiesis.

We detected slightly lower levels of the NET-associated peptides myeloperoxidase (MPO), neutrophil elastase, and S100A9 (a subunit of calprotectin, a metal-sequestering protein associated with neutrophils) in the cecal content supernatant of STm-infected *Ccl28*$^{-/-}$ mice compared to wild-type mice (*Figure 1—figure supplement 6*), though these differences were not statistically significant. Additionally, we quantified gut eosinophils, which commonly express the CCL28 receptor CCR3 (*Mohan et al., 2017*). Although the majority of eosinophils (CD11b$^+$ SiglecF$^+$ Side-scatter$^{High}$) detected in the gut and blood expressed CCR3 (*Figure 1—figure supplement 5B*), we found no alteration in their numbers in the gut, blood, or bone marrow in homeostasis or during STm infection (*Figure 1—figure*

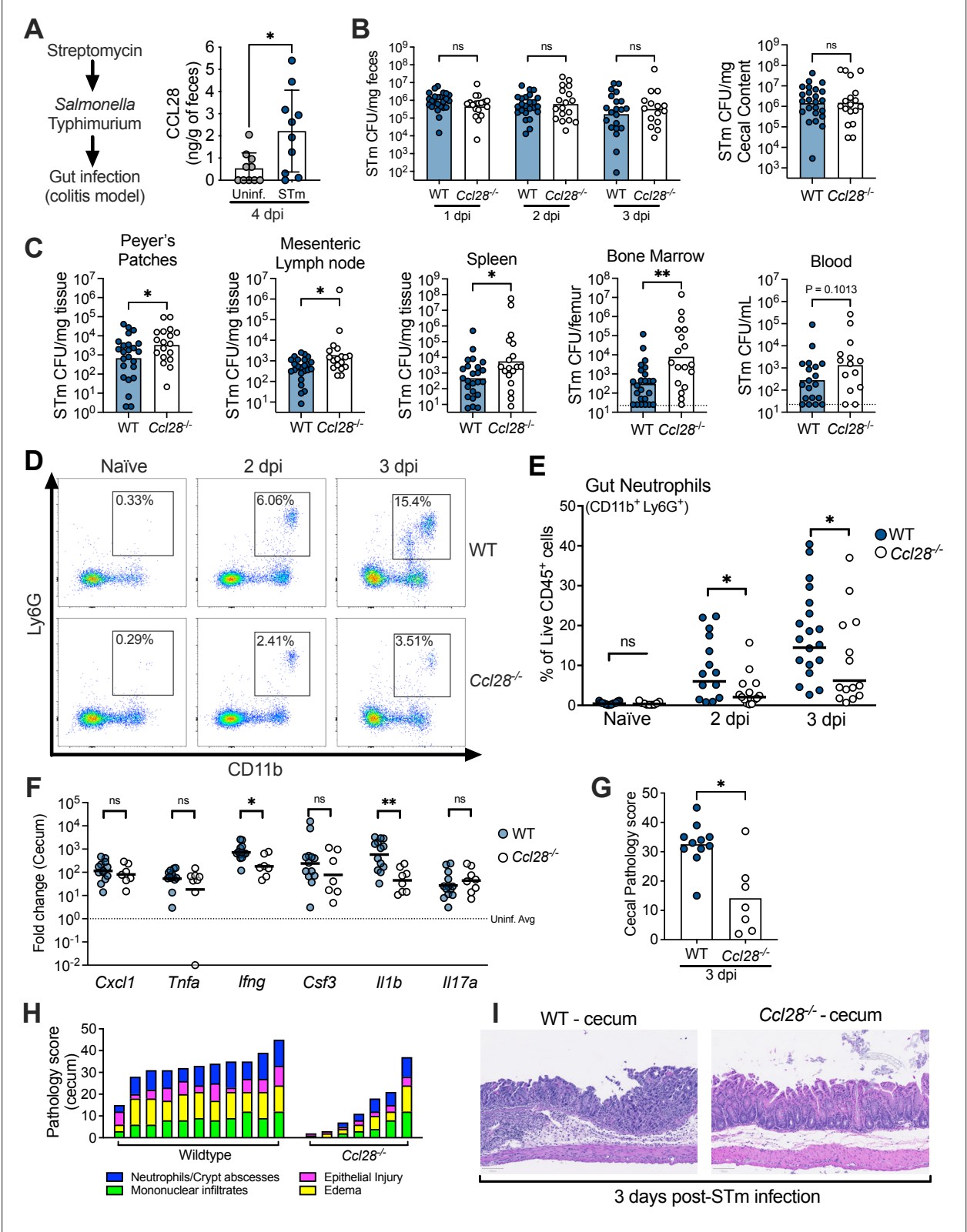

**Figure 1.** CCL28 confers protection during *Salmonella* colitis and promotes neutrophil accumulation in the gut. (**A**) For the colitis model, wild-type (WT) mice were gavaged with streptomycin 24 hr prior to oral infection with approximately 1 × 10⁹ CFU *S. enterica* serovar Typhimurium (STm). At 4 days post-infection (dpi), CCL28 in feces was quantified by ELISA. Data shown comprise two independent experiments (uninfected, *n* = 10; STm, *n* = 10). Bars represent the mean ± standard deviation (SD). (**B**) STm CFU in the fecal content collected 1–3 dpi, and in the cecal content 3 dpi from WT (filled circles)

*Figure 1 continued on next page*

*Figure 1 continued*

and *Ccl28$^{-/-}$* (white circles) littermate mice. (**C**) CFU recovered from the Peyer's patches, mesenteric lymph nodes, spleen, bone marrow, and blood at 3 dpi. Data shown comprise eight independent experiments (WT, *n* = 24; *Ccl28$^{-/-}$*, *n* = 18). Some of the spleen data points were published as a preliminary characterization in *Burkhardt et al., 2019* and are combined with the new dataset. Bars represent the geometric mean, dotted lines represent the limit of detection. (**D**) Representative pseudocolor dot plots of neutrophils (CD11b$^+$ Ly6G$^+$ cells; gated on live, CD45$^+$ cells) obtained from the gut tissues of uninfected (Naive) and STm-infected WT or *Ccl28$^{-/-}$* mice 2 or 3 dpi, as determined by flow cytometry. (**E**) Frequency of neutrophils in the live CD45$^+$ cells obtained from the gut mucosa of WT (filled circles) or *Ccl28$^{-/-}$* mice (white circles). Naive mouse data shown comprise four independent experiments (WT, *n* = 14; *Ccl28$^{-/-}$*, *n* = 9); 2 dpi data comprise four independent experiments (WT, *n* = 14; *Ccl28$^{-/-}$*, *n* = 14); 3 dpi data comprise eight independent experiments (WT, *n* = 24; *Ccl28$^{-/-}$*, *n* = 18). Bars represent the geometric mean. (**F**) Relative expression levels (qPCR) of *Cxcl1* (CXCL1), *Tnfa* (TNFα), *Ifng* (IFNγ), *Csf3* (G-CSF), *Il1b* (IL-1β), and *Il17a* (IL-17A) in the cecal tissue of STm-infected WT (filled circles, *n* = 13) or *Ccl28$^{-/-}$* mice (white circles, *n* = 8), 3 dpi, relative to uninfected control mice. Bars represent the geometric mean. Data shown comprise four independent experiments. (**G–I**) Histopathological analysis of the cecum collected from STm-infected WT or *Ccl28$^{-/-}$* mice, 3 dpi (WT, *n* = 11; *Ccl28$^{-/-}$*, *n* = 7). Scale bars indicate 100 μm. (**G**) Sum of the total histopathology score (bars represent the mean; symbols represent individual mice), (**H**) histopathology scores showing the individual analyzed parameters of each mouse (stacked bar height represents the overall score), and (**I**) hematoxylin and eosin (H&E)-stained sections from one representative animal for each group (×200 magnification). For (**B**) and (**C**), CFU data were log-normalized before statistical analysis by Welch's *t* test. Mann–Whitney *U* was used for all other datasets where statistical analysis was performed. A significant difference relative to WT controls is indicated by *p ≤ 0.05, **p ≤ 0.01; ns, not significant.

The online version of this article includes the following figure supplement(s) for figure 1:

**Figure supplement 1.** *Salmonella* gut colonization and extraintestinal levels 2 days post-infection.

**Figure supplement 2.** CCL28 does not confer protection in a *Salmonella* bacteremia model, and lacks direct antimicrobial activity against *Salmonella*.

**Figure supplement 3.** Flow cytometry gating strategy for the identification and classification of major immune cell populations in the tissues of STm-infected mice.

**Figure supplement 4.** Wild-type (WT) and *Ccl28$^{-/-}$* mice exhibit similar numbers of B and T cells in the gut, blood, and bone marrow.

**Figure supplement 5.** Profiling granulocyte and APC-like cell abundance in wild-type (WT) and *Ccl28$^{-/-}$* mouse tissues during STm infection.

**Figure supplement 6.** Neutrophil-associated antimicrobial protein levels during intestinal STm infection of wild-type (WT) and *Ccl28$^{-/-}$* mice.

*supplement 5C*). The abundance of other innate immune cell populations (CD11b$^+$ CD11c$^+$ conventional dendritic cell-like cells and CD11b$^+$ F4/80$^+$ macrophage-like cells) responding to STm in the gut also showed no major differences (*Figure 1—figure supplement 5D, E*). Therefore, CCL28 specifically promotes neutrophil accumulation in the gut during STm infection, which occurs after neutrophil production in the bone marrow and their egress into the blood circulation.

## Gut proinflammatory gene expression and tissue pathology are reduced in *Ccl28$^{-/-}$* mice infected with STm

Neutrophils can mediate inflammation by producing proinflammatory molecules or engaging in cross-talk with other cells (*Sabroe et al., 2005*). We evaluated the expression of genes encoding proinflammatory cytokines in the cecum of *Ccl28$^{-/-}$* mice and wild-type littermates 3 dpi with STm. *Ifng* and *IL1b* gene transcripts were significantly higher in the cecum of infected wild-type mice compared to *Ccl28$^{-/-}$* mice, while other factors involved in neutrophil recruitment (*Cxcl1*, *Csf3*, and *Il17a*) or the proinflammatory cytokine *Tnfa* showed no significant differences (*Figure 1F*). No differences were observed between uninfected wild-type mice and *Ccl28$^{-/-}$* mice (data not shown). Histopathology at 3 dpi revealed marked cecal inflammation, including significant neutrophil recruitment in wild-type mice, which was greatly reduced in *Ccl28$^{-/-}$* mice (*Figure 1G–I*). Thus, CCL28 modulates neutrophil accumulation and drives inflammatory tissue pathology and colitis during STm infection.

## *Ccl28$^{-/-}$* mice are protected from lethal infection in an *Acinetobacter* pneumonia model

CCL28 is expressed in several mucosal tissues beyond the gut, including the lung (*Mohan et al., 2017*). To investigate whether CCL28 promotes neutrophil accumulation and host protection in the lung, we employed a murine Ab pneumonia model (*Dillon et al., 2019*; *Lin et al., 2015*). Ab is an emerging, frequently multidrug-resistant Gram-negative pathogen causing potentially lethal nosocomial pneumonia (*Ayoub Moubareck and Hammoudi Halat, 2020*). Following intratracheal Ab infection, we observed a striking phenotype: 75% of wild-type mice died within 48 hr, whereas 88% of *Ccl28$^{-/-}$* knockout mice survived through 10 dpi (*Figure 2A*). The enhanced resistance of *Ccl28$^{-/-}$* mice

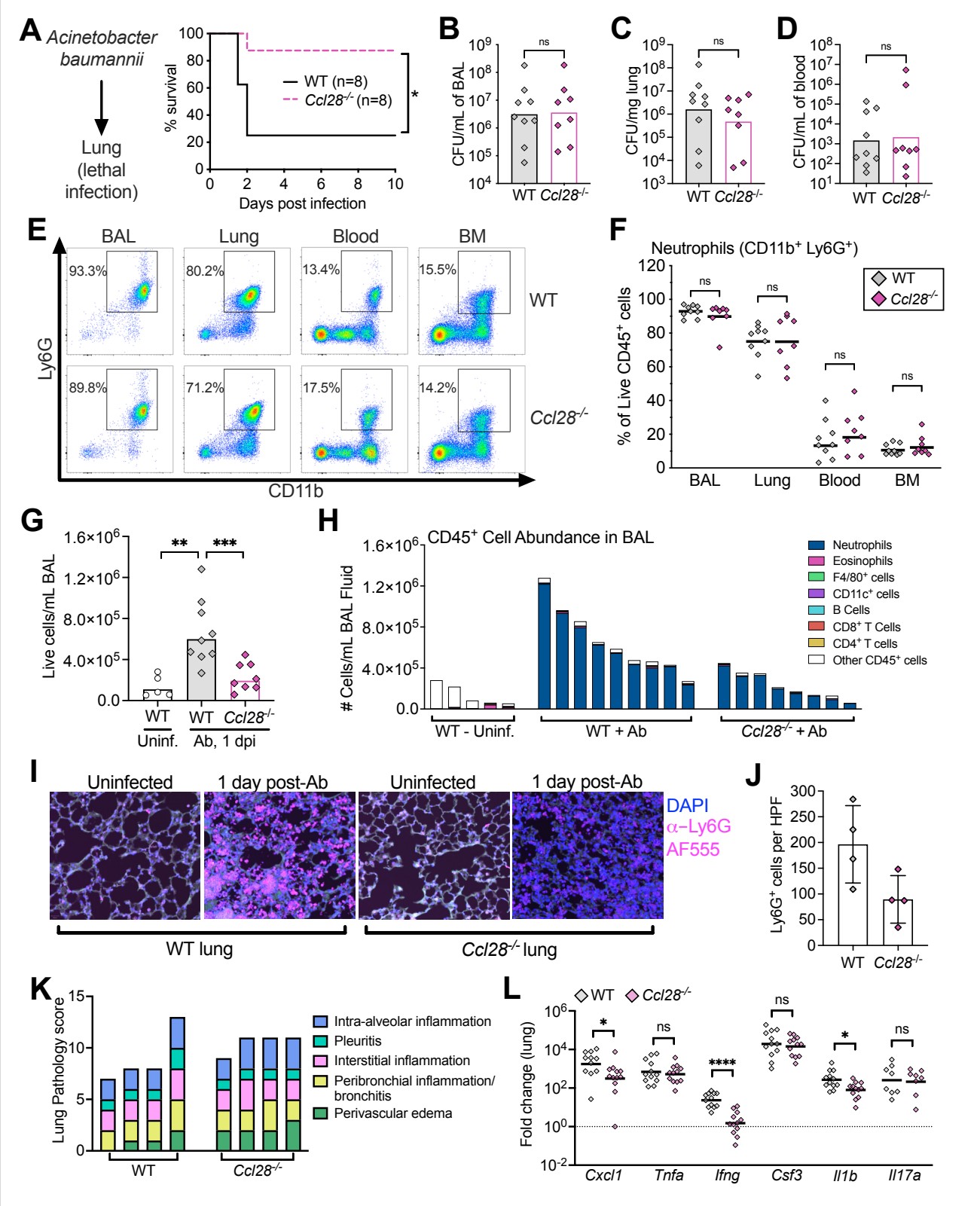

**Figure 2.** Absence of CCL28 confers protection in a lethal *Acinetobacter* pneumonia model. (**A**) Wild-type (WT) mice (solid black line) and *Ccl28⁻/⁻* mice (dashed magenta line) were intratracheally infected with approximately 1 × 10⁸ CFU *Acinetobacter baumannii* (Ab) and their survival was determined for 10 days. Data shown comprise two independent experiments (WT, *n* = 8; *Ccl28⁻/⁻*, *n* = 8). (**B–H**) WT mice (*n* = 9) and *Ccl28⁻/⁻* mice (*n* = 8) were intratracheally infected with Ab and sacrificed 1 day post-infection (dpi). Data shown comprise three independent experiments. Symbols represent data

*Figure 2 continued on next page*

*Figure 2 continued*

from individual mice. (**B–D**) Ab CFU were quantified from the BAL (bronchoalveolar lavage) fluid, (**C**) lung tissue, and (**D**) blood in WT (gray symbols) and *Ccl28*⁻ᐟ⁻ mice (magenta symbols). Bars represent the geometric mean. (**E**) Representative pseudocolor dot plots of neutrophils (CD11b⁺ Ly6G⁺ cells; gated on live, CD45⁺ cells) and (**F**) frequency of neutrophils obtained from the BAL, lung, blood, and bone marrow of Ab-infected WT or *Ccl28*⁻ᐟ⁻ mice, as determined by flow cytometry. Lines represent the geometric mean. (**G**) The number of live host cells per mL of BAL, determined using an automated cell counter with Trypan Blue counterstain to assess viability, from uninfected WT (Uninf., *n* = 5), and Ab-infected WT (*n* = 9); and *Ccl28*⁻ᐟ⁻ mice (*n* = 8). Bars represent the geometric mean. (**H**) Relative abundance of different leukocyte populations as a proportion of the live CD45⁺ cell population was assessed in the BAL. Each bar represents data from one mouse. (**I**) Representative immunofluorescence image of lungs from WT and *Ccl28*⁻ᐟ⁻ mice, uninfected or infected with Ab, stained for the neutrophil marker Ly6G (magenta). 4′,6-diamidino-2-phenylindole (DAPI, blue) was used to label nuclei. Scale bars indicate 20 μm. (**J**) Quantification of Ly6G⁺ cells per high-power field (HPF) from immunofluorescence images of lungs from WT mice (*n* = 4) and *Ccl28*⁻ᐟ⁻ mice (*n* = 4). Bars represent the mean ± standard deviation (SD). (**K**) Histopathological analysis of lungs from WT and *Ccl28*⁻ᐟ⁻ mice infected with Ab at 1 dpi. Each bar represents an individual mouse. (**L**) Relative expression levels (qPCR) of *Cxcl1* (CXCL1), *Tnfa* (TNFα), *Ifng* (IFNγ), *Csf3* (G-CSF), *Il1b* (IL-1β), and *Il17a* (IL-17A) in the lung of WT (*n* = 11) or *Ccl28*⁻ᐟ⁻ mice (*n* = 12) infected with Ab (1 dpi). Bars represent the geometric mean. Data shown comprise three independent experiments. For (**A**), survival curves were statistically compared using a log-rank (Mantel–Cox) test. For (**B–D**), CFU data were log-normalized before analysis by Welch's *t* test. For (**F**), (**G**), and (**L**), Mann–Whitney *U* was used to compare groups with unknown distribution. A significant difference between groups is indicated by *p ≤ 0.05, **p ≤ 0.01, ***p ≤ 0.001, ****p ≤ 0.0001. ns, not significant.

The online version of this article includes the following figure supplement(s) for figure 2:

**Figure supplement 1.** Immunophenotyping of CD11b⁺ immune cells recovered from wild-type (WT) and *Ccl28*⁻ᐟ⁻ mice during *A. baumannii* infection.

**Figure supplement 2.** Immunophenotyping of lymphocytes recovered from wild-type (WT) and *Ccl28*⁻ᐟ⁻ mice during *A. baumannii* infection.

**Figure supplement 3.** Neutrophil-associated antimicrobial protein levels during lung Ab infection of wild-type (WT) and *Ccl28*⁻ᐟ⁻ mice.

was not associated with significant reductions in Ab CFU recovered at 1 dpi from bronchoalveolar lavage (BAL) fluid, lung, or blood (*Figure 2B–D*). These results suggest that, unlike STm gut infection, CCL28 exacerbates lethality during Ab lung infection.

In vitro, high concentrations (1 μM) of CCL28 exhibited direct antimicrobial activity against 5 × 10⁵ CFU of Ab, but not when higher CFU (5 × 10⁸/ml) were used as inoculum in the assay (*Figure 1—figure supplement 2C*). Given that high Ab CFU were recovered in the lung of wild-type mice (*Figure 2B, C*), CCL28 does not appear to limit growth of this pathogen in vivo even though it exhibits modest antimicrobial activity in vitro. We thus investigated if alterations in neutrophil accumulation in the lung between wild-type and *Ccl28*⁻ᐟ⁻ mice could explain the higher lethality of *Ccl28*⁻ᐟ⁻ mice challenged with Ab lung infection.

## CCL28 promotes neutrophil accumulation to the lung during *Acinetobacter* infection

Prior studies demonstrated neutrophil recruitment to the lungs of Ab-infected mice beginning at 4 hr post-infection and peaking at 1 dpi (*van Faassen et al., 2007*; *Tsuchiya et al., 2012*). CCL28 contributed to neutrophil recruitment during STm gut infection, so we analyzed neutrophil recruitment to the lung mucosa 1 day after Ab infection in wild-type and *Ccl28*⁻ᐟ⁻ mice. Neutrophils (CD11b⁺ Ly6G⁺) were the majority of immune cells in the BAL fluid and lungs of both wild-type and *Ccl28*⁻ᐟ⁻ mice (*Figure 2E, F*). However, greater cellular infiltrates were recovered in the BAL fluid of wild-type mice compared to *Ccl28*⁻ᐟ⁻ littermates (*Figure 2G*). Neutrophils made up the majority of BAL cells in all Ab-infected mice, but were less abundant in *Ccl28*⁻ᐟ⁻ mice (*Figure 2H*), while neutrophil percentages in lung tissues, and neutrophil numbers in the blood or bone marrow, did not differ significantly between the wild-type and mutant mice (*Figure 2F*). Although neutrophil abundance greatly increased in the lungs during Ab infection (*Figure 2—figure supplement 1A*), no other cell types profiled varied between wild-type and *Ccl28*⁻ᐟ⁻ mice before or 1 day post-Ab infection (*Figure 2—figure supplement 1B–D* and *Figure 2—figure supplement 2A–C*), besides a slight deficiency in lung eosinophil levels in uninfected *Ccl28*⁻ᐟ⁻ mice (*Figure 2—figure supplement 1B*). Although substantial lung inflammation was observed in both wild-type and *Ccl28*⁻ᐟ⁻ mice post-infection (*Figure 2I, K*), immunofluorescence analysis revealed fewer neutrophils (Ly6G⁺ cells) in the lungs of *Ccl28*⁻ᐟ⁻ mice (*Figure 2I, J*). Levels of elastase, MPO, and S100A9 in the BAL fluid supernatant were higher in Ab-infected mice compared to uninfected controls, with a trend toward lower levels in *Ccl28*⁻ᐟ⁻ mice (*Figure 2—figure supplement 3*). Gene expression of IFNγ and IL-1β was significantly lower in Ab-infected lungs of *Ccl28*⁻ᐟ⁻ mice compared to wild-type mice (*Figure 2L*), while *Cxcl1* gene expression was reduced and the other proinflammatory genes tested (*Il17a*, *Csf3*, *Tnfa*) did not differ (*Figure 2L*). Therefore, CCL28

contributes to lung inflammation and neutrophil accumulation during Ab pneumonia, similar to its role in STm gut infection.

## Gut and BAL neutrophils express receptors CCR3 and CCR10 during infection

CCL28 attracts leukocytes expressing at least one of its receptors, CCR3 or CCR10. CCR10 is found on T cells, B cells, and IgA-secreting plasma cells, whereas eosinophils express CCR3 (*Mohan et al., 2017*). Although early studies concluded that CCR3 was absent in neutrophils (*Höchstetter et al., 2000*), later research detected this receptor on neutrophils isolated from patients with chronic inflammation (*Hartl et al., 2008*). Based on these findings and our observations of CCL28-dependent neutrophil accumulation in the gut during STm colitis and in the lung during Ab infection (*Figures 1 and 2*), we performed flow cytometry on single-cell suspensions from infected mouse tissues to evaluate surface expression of CCR3 and CCR10. In STm-infected mice, we analyzed the gut, blood, and bone marrow (*Figure 3A, B*). Both receptors were present on a small subset of bone marrow neutrophils (~4% CCR3, ~0.2% CCR10) and blood neutrophils (~5% CCR3, ~1% CCR10) during infection. However, neutrophils expressing these receptors, particularly CCR3, were enriched in the inflamed gut, with ~20% expressing CCR3 and ~2% expressing CCR10 (*Figure 3A, B*). Simultaneously staining for both CCR3 and CCR10 showed that ~1% of gut neutrophils from infected wild-type mice expressed both receptors (*Figure 3—figure supplement 1A*), and infected *Ccl28$^{-/-}$* mice expressed similar levels of these receptors as wild-type mice (*Figure 3—figure supplement 1B*).

Neutrophils isolated from the BAL of Ab-infected wild-type mice also expressed CCR3 and CCR10 surface expression, with ~15% of neutrophils expressing CCR3 (*Figure 3C*) and ~2% expressing CCR10 (*Figure 3D*). Simultaneously staining for both CCR3 and CCR10 revealed that ~0.5% of BAL neutrophils from infected wild-type mice expressed both receptors (*Figure 3—figure supplement 1C*), and infected *Ccl28$^{-/-}$* mice expressed similar levels of these receptors as wild-type mice (*Figure 3—figure supplement 1D*). Surprisingly, a similar percentage of neutrophils isolated from the blood and the bone marrow of Ab-infected mice expressed these receptors compared to BAL neutrophils (*Figure 3C, D*). These findings suggest that CCR3 and CCR10 expression is higher in neutrophils associated with mucosal tissues, potentially facilitating their accumulation in these tissues or being induced upon recruitment to the mucosal sites.

## Proinflammatory stimuli and phagocytosis induce expression of CCR3 and CCR10 on neutrophils

We investigated mechanisms underpinning the upregulation of CCR3 and CCR10 in neutrophils. A prior study indicated that a cocktail of proinflammatory cytokines (GM-CSF, IFNγ, TNFα) boosts CCR3 expression in human peripheral blood neutrophils from healthy donors (*Hartl et al., 2008*), and expression of these cytokines is highly induced during STm colitis (*Figure 1F*) and Ab pneumonia (*Figure 2L*). We stimulated bone marrow neutrophils from wild-type mice (which express low levels of CCR3 and CCR10) with these cytokines, and independently with other pro-inflammatory compounds including lipopolysaccharide (LPS), the protein kinase C activator phorbol 12-myristate 13-acetate (PMA), or the *N*-formylated, bacterial-derived chemotactic peptide fMLP. PMA produced the highest expression of CCR3 (~30% CCR3$^+$ neutrophils, 10-fold induction compared to baseline), while the GM-CSF + IFNγ + TNFα cytokine combination or fMLP induced moderate CCR3 expression (~15% CCR3$^+$, a fivefold increase) and LPS yielding the lowest but still significant induction (~10% CCR3$^+$, a threefold increase) (*Figure 3E*). Trends in CCR10 expression were similar to CCR3, though no stimuli induced more than ~0.5% CCR10$^+$ neutrophils (~1.2- to 2.5-fold higher than baseline) (*Figure 3F*).

Phagocytosis of microbes and necrotic debris are critical neutrophil functions at tissue foci of infection and inflammation (*Uribe-Querol and Rosales, 2020*) and are associated with changes in neutrophil gene expression (*Kobayashi et al., 2002*). We tested whether phagocytosis induced CCR3 and CCR10 expression by incubating bone marrow neutrophils with latex beads, with or without the cytokine cocktail. Phagocytosis of latex beads alone resulted in a small but significant induction of neutrophil CCR3 expression (~8% of neutrophils); however, latex beads augmented with the cytokine cocktail markedly induced CCR3 expression (~25% of neutrophils vs. ~15% with cocktail alone; *Figure 3G*). This synergistic effect of phagocytosis was not notable for CCR10 (*Figure 3H*).

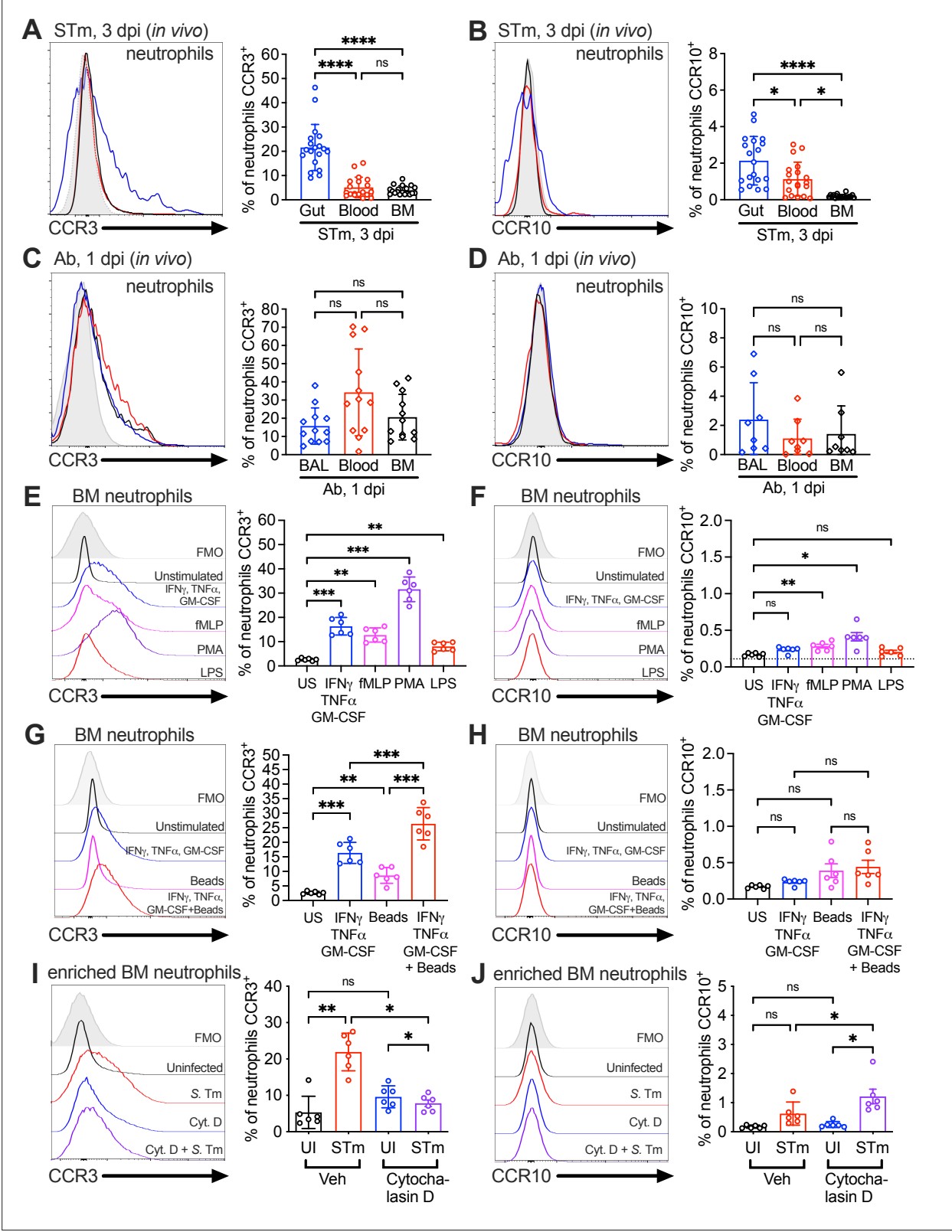

**Figure 3.** Surface expression of the CCL28 receptors CCR3 and CCR10 on neutrophils from infected tissue, and upon stimulation with proinflammatory stimuli and phagocytosis. Surface expression of (**A, C**) CCR3 or (**B, D**) CCR10 on murine neutrophils obtained from (**A, B**) the gut, blood, and bone marrow (BM) 3 dpi with STm, or (**C, D**) the bronchoalveolar lavage (BAL), blood, and bone marrow 1 dpi with Ab, analyzed by flow cytometry. Left panels show representative histograms of (**A, C**) CCR3 or (**B, D**) CCR10 expression on the surface of neutrophils (gated on live, CD45⁺ CD11b⁺ Ly6G⁺

*Figure 3 continued on next page*

*Figure 3 continued*

cells) from (**A, B**) the gut (blue), blood (red), and bone marrow (BM; black) or (**C, D**) BAL (blue), blood (red), and bone marrow (BM; black). Right panels show the percentage of (**A, C**) CCR3$^+$ or (**B, D**) CCR10$^+$ neutrophils obtained from (**A, B**) gut, blood, and BM or (**C, D**) BAL, blood, and BM. Data are from six independent experiments. (**E–H**) Uninfected bone marrow neutrophils were unstimulated or treated with the indicated stimuli for 4 hr. Surface expression of (**E, G**) CCR3 and (**F, H**) CCR10 on neutrophils was determined by flow cytometry. Left panels show representative histograms of (**E, G**) CCR3 or (**F, H**) CCR10 surface expression after stimulation with: (**E, F**) cytokines IFNγ + TNFα + GM-CSF (blue); fMLP (magenta); phorbol 12-myristate 13-acetate (PMA) (purple); lipopolysaccharide (LPS) (red); (**G, H**) cytokines IFNγ + TNFα + Granulocyte-macrophage colony stimulating factor (GM-CSF, blue); beads alone (magenta); cytokines plus beads (red). Right panels show the percentage of (**E, G**) CCR3$^+$ or (**F, H**) CCR10$^+$ neutrophils following stimulation with the indicated stimuli. US = unstimulated. Data shown are pooled from two independent experiments. (**I, J**) Bone marrow cells enriched for neutrophils were infected with opsonized STm at a multiplicity of infection (MOI) = 10 for 1 hr with (violet) or without (red) pretreatment with cytochalasin D for 30 min before infection. Surface expression of (**I**) CCR3 or (**J**) CCR10 was determined by flow cytometry. Data are from two independent experiments. Left panels show representative histograms of surface receptor staining on neutrophils, and right panels show the percentages. (**A–J**, right panels) Bars represent the mean ± standard deviation (SD). (**A–D**) Data were analyzed by one-way analysis of variance (ANOVA) for paired samples (non-parametric Friedman test), assuming non-normal distribution and non-equal SD given the differences in the variance among the groups, followed by Dunn's multiple comparisons test. (**E–J**) Data were analyzed by one-way ANOVA for paired samples, applying the Greenhouse–Geisser correction given the differences in variance among the groups; Bonferroni's multiple comparison test was performed to compare between relevant stimulation conditions. Significant changes are indicated by *p ≤ 0.05, **p ≤ 0.01, ***p ≤ 0.001, ****p ≤ 0.0001; ns, not significant.

The online version of this article includes the following figure supplement(s) for figure 3:

**Figure supplement 1.** Expression of CCR3 and CCR10 in neutrophils isolated from the gut and lung mucosa in infected wild-type (WT) and *Ccl28*$^{-/-}$ mice.

To further probe the role of phagocytosis in CCR3 expression, we incubated bone marrow neutrophils with live STm for 1 hr. STm rapidly induced CCR3 expression on the neutrophil surface (~25% of cells; *Figure 3I*), whereas CCR10 was only minimally induced (*Figure 3J*). Cytochalasin D, a potent inhibitor of the actin polymerization required for phagocytic uptake, largely blocked CCR3 receptor induction (*Figure 3I*); however, CCR10 induction was not blocked (*Figure 3J*), suggesting that a mechanism other than phagocytic uptake likely drives the minor increase in CCR10 expression by neutrophils. Incubation of bone marrow neutrophils with CCL28 (both alone and in the context of STm co-incubation) had negligible effects on CCR3 and CCR10 levels (data not shown). Thus, proinflammatory stimuli and phagocytosis enhance CCR3 and, to a lesser extent, CCR10 expression on the neutrophil surface.

## CCR3 is stored intracellularly in neutrophils

Neutrophil intracellular compartments and granules harbor enzymes, cytokines, and receptors necessary for rapid responses to pathogens. For example, activation of human neutrophils induces rapid translocation of complement receptor type 1 (CR1) from an intracellular compartment to the cell surface, increasing its surface expression up to 10-fold (*Berger et al., 1991*). Given the rapid (within 1 hr) increase of neutrophil CCR3 surface expression upon STm infection, we hypothesized that CCR3, akin to CR1, may be stored intracellularly in neutrophils, consistent with reports of intracellular CCR3 in eosinophils (*Spencer et al., 2006*).

Uninfected bone marrow neutrophils maintained relatively low surface levels of CCR3 (*Figure 4A*), but when permeabilized for intracellular staining, almost all (~99%) were CCR3$^+$, indicating intracellular storage (*Figure 4B*). Upon STm infection in vitro, bone marrow neutrophils increased CCR3 surface expression as quickly as 5 min post-infection, reaching a maximum of ~30% CCR3$^+$ neutrophils at 2 hpi (*Figure 4A*). These results suggest mobilization of pre-formed receptor from an intracellular compartment (*Figure 4B*). Intracellular stores of CCR10 were also detected in some bone marrow neutrophils under homeostatic conditions, with a small but significant increase during STm infection (*Figure 4—figure supplement 1B*). However, CCR10 was expressed on the surface of only ~0.3% uninfected bone marrow neutrophils, increasing to ~0.6% during STm infection (*Figure 4—figure supplement 1A*). In vitro, Ab infection induced less CCR3 surface expression on neutrophils relative to STm (~7–10%) and took longer to observe the increased CCR3$^+$ staining (*Figure 4C*), whereas CCR10 did not significantly increase (*Figure 4—figure supplement 1C*). Most bone marrow neutrophils also expressed intracellular CCR3 (*Figure 4D*) and CCR10 (*Figure 4—figure supplement 1D*) during Ab infection. Similar findings were observed in neutrophils isolated from bone marrow, blood, and gut tissue of mice orally infected with STm, and from bone marrow, blood, and BAL fluid of mice infected with Ab, with both intracellular and surface CCR3 observed (*Figure 4E, F*). CCR3 surface expression

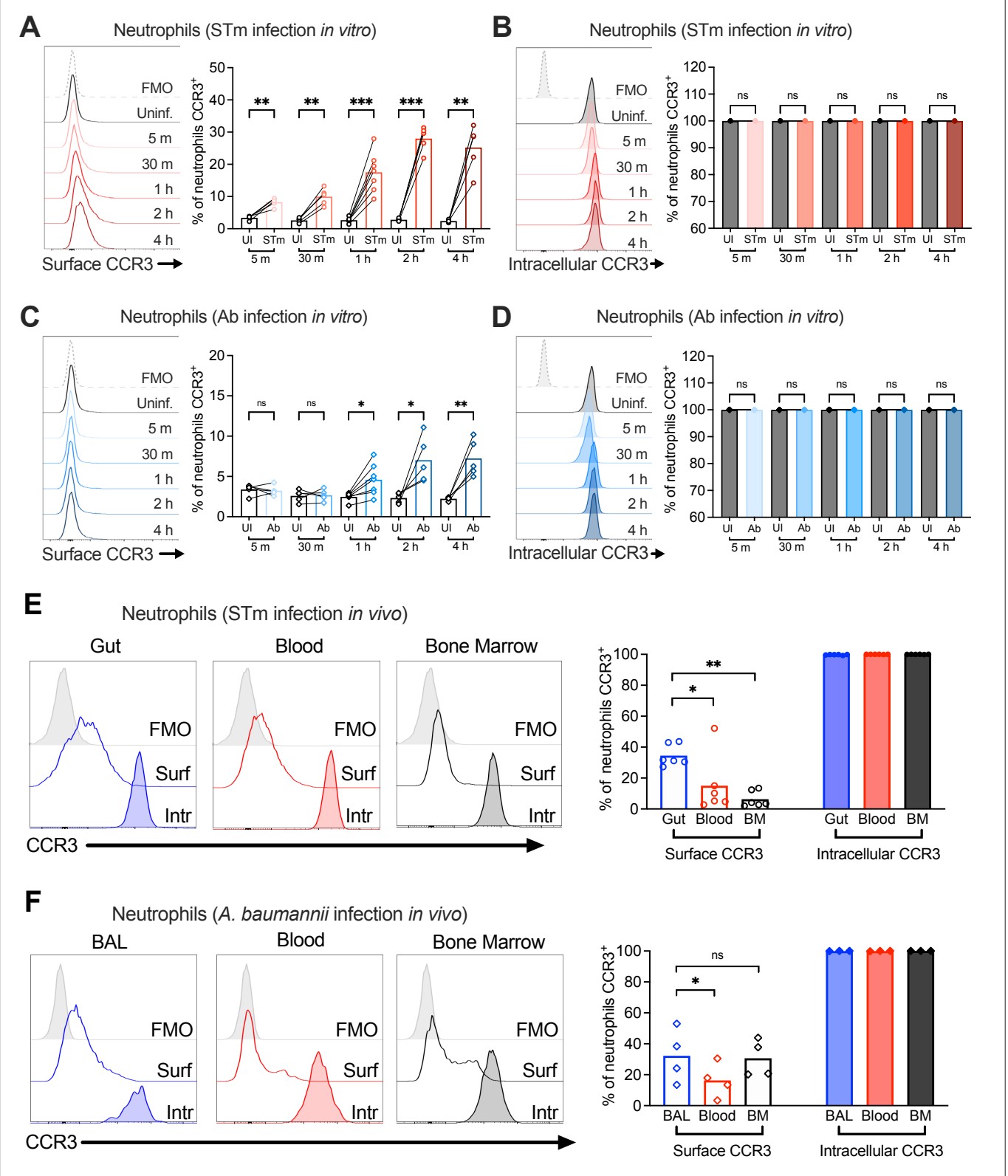

**Figure 4.** Neutrophil CCR3 is stored in intracellular compartments and rapidly mobilizes to the cell surface during infection. Neutrophils enriched from wild-type mouse bone marrow were infected at multiplicity of infection (MOI) = 10 for 5 min to 4 hr with (**A, B**) opsonized *Salmonella enterica* serovar Typhimurium (STm) or (**C, D**) *Acinetobacter baumannii* (Ab). (**A, C**) Surface CCR3 or (**B, D**) intracellular CCR3 staining was detected by flow cytometry. Connected symbols represent data from neutrophils collected from the same mouse under different stimulation conditions. Neutrophils were obtained

*Figure 4 continued on next page*

*Figure 4 continued*

from (**E**) the gut, blood, and bone marrow 3 dpi with STm or (**F**) bronchoalveolar lavage (BAL), blood, and bone marrow 1 dpi with Ab. Surface (clear histograms) or intracellular (filled histograms) CCR3 expression was analyzed by flow cytometry. (**A–F**) Left panels show representative histograms, and right panels show the percentage of neutrophils expressing CCR3 on their surface (clear bars) or intracellularly (filled bars). Bars represent the mean. Data were analyzed by paired *t* test (**A–D**) or one-way analysis of variance (ANOVA) followed by Tukey's multiple comparison test (**E, F**) on log-transformed data. Significant changes are indicated by *p ≤ 0.05, **p ≤ 0.01, ***p ≤ 0.001; ns, not significant.

The online version of this article includes the following figure supplement(s) for figure 4:

**Figure supplement 1.** Expression kinetics of neutrophil CCR10.

levels were higher on neutrophils isolated from the gut relative to other sites (*Figure 4E*), though levels in the BAL fluid were similar to Ab-infected blood and bone marrow neutrophils (*Figure 4F*). Neutrophils expressing surface CCR10 were low in all tissues, though slightly higher in the STm-infected gut than in blood and bone marrow, with intracellular stores of CCR10 also observed (*Figure 4—figure supplement 1E, F*). We conclude that CCR3 is stored intracellularly in neutrophils and rapidly mobilized to the cell surface upon infection, phagocytosis, and/or cytokine stimulation.

## CCL28 enhances neutrophil antimicrobial activity, ROS production, and NET formation via CCR3 stimulation

Chemokines are essential for neutrophil migration to infection sites and may regulate additional neutrophil bactericidal effector functions, including the production of ROS and formation of NETs (*Capucetti et al., 2020*). We tested if CCL28 has chemotactic and/or immunostimulatory activity toward bone marrow neutrophils in vitro after boosting their CCR3 surface expression with the cytokine cocktail (GM-CSF + IFNγ + TNFα) as shown in *Figure 3*. We incubated the neutrophils with CCL28, the well-known neutrophil chemoattractant CXCL1, or with CCL11/eotaxin, a chemokine that binds CCR3 and is induced in the asthmatic lung to promote eosinophil recruitment (*Conroy and Williams, 2001*; *Garcia-Zepeda et al., 1996*; *Kitaura et al., 1996*). We found that CCL28 promoted neutrophil chemotaxis, though not as potently as CXCL1, while CCL11 had no significant effect (*Figure 5A*).

To test whether CCL28 stimulation enhanced neutrophil effector function, we incubated STm with bone marrow neutrophils for 2.5 hr with or without CCL28 (50 nM) or CCL11 (50 nM), then quantified bacterial killing. Stimulation with CCL28 significantly increased neutrophil bactericidal activity against STm, with ~40% of the bacterial inoculum cleared, compared to ~10% clearance by unstimulated neutrophils (*Figure 5B*). Neutrophils stimulated with CCL11 displayed an intermediate phenotype (~25% bacterial killing). Neither chemokine exhibited direct antimicrobial activity against STm (*Figure 1—figure supplement 2D*). In contrast, ex vivo neutrophil killing of Ab was not significantly enhanced by CCL28 or CCL11 treatment (*Figure 5C*). Thus, although CCL28 modulates neutrophil accumulation in the lung during Ab infection (*Figure 2D–J*), it fails to reduce pathogen burden in the lung (*Figure 2B*) likely because CCL28 stimulation does not enhance neutrophil bactericidal activity against Ab.

Our data indicate that CCR3 is the primary CCL28 receptor expressed in neutrophils during STm infection (*Figure 3I and 4*). We tested whether the CCL28-mediated increase in neutrophil bactericidal activity could be reversed using SB328437, a CCR3 antagonist (*White et al., 2000*). SB328437 reversed the effects of both CCL28 and CCL11 on neutrophils, confirming receptor specificity (*Figure 5D*). An important mechanism of bacterial killing is the production of ROS (*Fang, 2011*), which is triggered by infection and enhanced by proinflammatory stimuli including cytokines and chemokines (*Nguyen et al., 2017*). We measured ROS production by incubating neutrophils with the cell-permeable probe 2',7'-dichlorodihydrofluorescein diacetate (H$_2$DCFDA), which forms the fluorescent byproduct 2',7'-dichorofluorescein (DCF) when oxidized by ROS, and found that CCL28 stimulation enhanced neutrophil ROS production during STm infection (*Figure 5E*). The increased ROS production triggered by CCL28 was reversed when neutrophils were incubated with an anti-CCR3 blocking antibody (*Figure 5F*), but not with an anti-CCR10 blocking antibody (*Figure 5G*).

In addition to their direct antimicrobial activity, ROS trigger other neutrophil responses, including NET formation (*Nguyen et al., 2017*). NETs can be induced by various stimuli, including microbial products, inflammatory cytokines and chemokines, immune complexes, and activated platelets (*Boeltz et al., 2019*). To determine whether CCL28 enhances NET formation, we incubated human neutrophils with activated platelets with or without CCL28, then incubated the cells with the DNA-staining

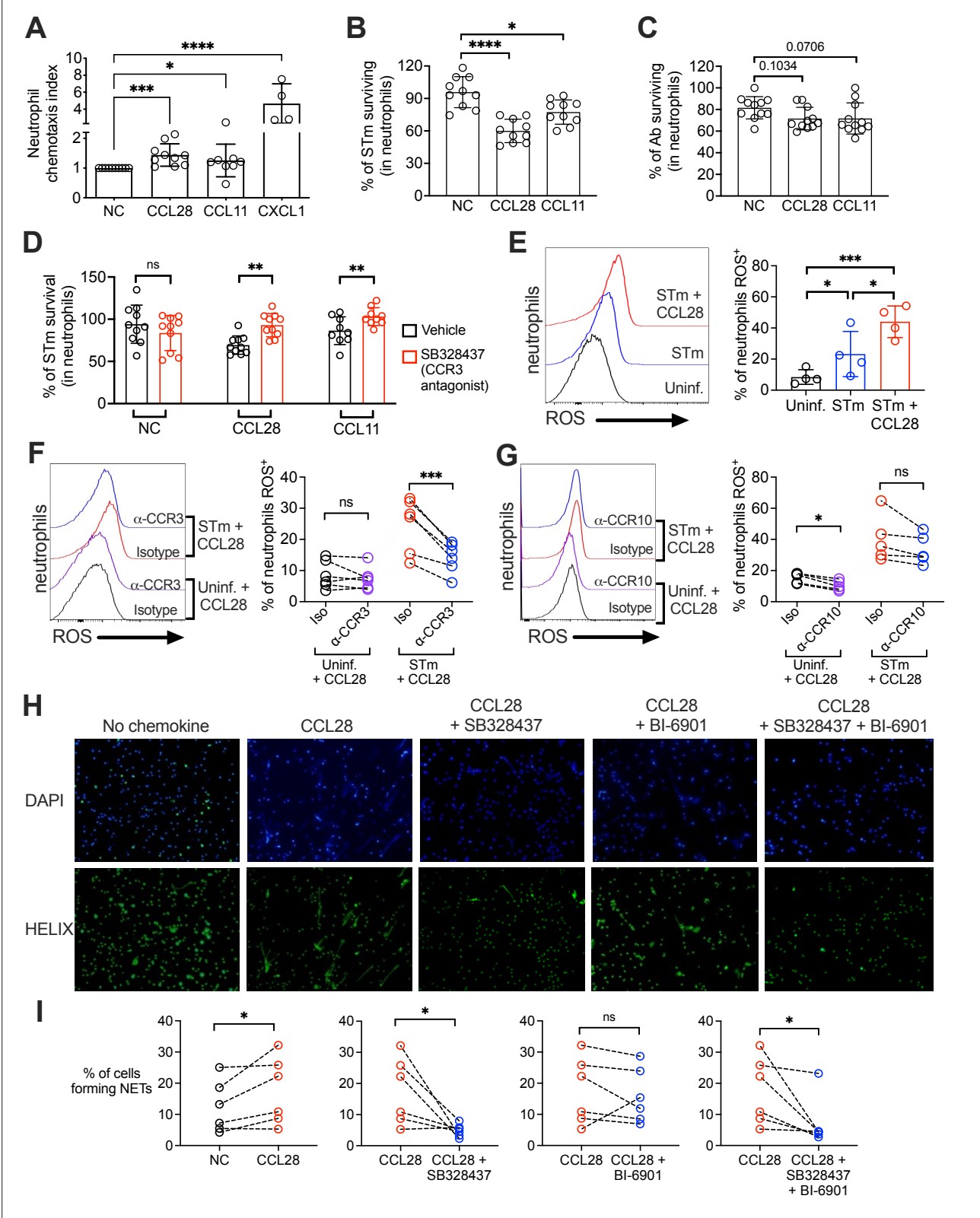

**Figure 5.** CCL28 enhances neutrophil antimicrobial activity. (**A**) Murine bone marrow neutrophils were stimulated with IFNγ + TNFα + GM-CSF for 4 hr before adding 1 × 10⁶ cells to the upper compartment of a transwell chamber for chemotaxis assays. Each of the chemokines (CCL28, CCL11, or CXCL1), or no chemokine (NC), was placed in separate lower compartments. The transwell plate was incubated for 2 hr at 37°C. Cells that migrated to the lower compartment were enumerated by flow cytometry. Neutrophil chemotaxis index was calculated by taking the number of cells that migrated in response

*Figure 5 continued on next page*

*Figure 5 continued*

to a chemokine and dividing it by the number of cells that migrated in the absence of a chemokine. Data are from four independent experiments. (**B, C**) Infection of bone marrow neutrophils. (**B**) Opsonized STm (1 × 10$^7$ CFU) or (**C**) opsonized Ab (1 × 10$^7$ CFU) were cultured alone, or added to bone marrow neutrophils (1 × 10$^6$ cells) stimulated with CCL28, CCL11, or no chemokine, for 2.5 hr (STm) or 4.5 hr (Ab) at 37°C. Neutrophils were lysed with 1% Triton-X and surviving bacteria were enumerated by plating serial dilutions. Percentage of bacterial survival was calculated for each condition by taking the CFU from bacteria incubated with neutrophils and dividing it by the CFU from bacteria incubated without neutrophils, multiplied by 100. Data shown for each infection comprise three independent experiments. Bars represent the mean ± standard deviation (SD). (**D**) The effect of the CCR3 antagonist SB328437 on neutrophil-mediated STm killing was evaluated by performing the experiment as described in panel (**B**), with or without the antagonist. Data shown comprise three independent experiments. (**E–G**) Reactive oxygen species (ROS) production (2',7'-dichlorodihydrofluorescein diacetate [H$_2$DCFDA] conversion to fluorescent DCF) detected by flow cytometry in bone marrow neutrophils infected with STm as described in panel (**B**). In (**F, G**), cells were stimulated with CCL28 in the presence of an anti-CCR3 antibody, an anti-CCR10 antibody, or isotype controls. Left panels show representative histograms, and right panels show the percentage of ROS$^+$ neutrophils in the indicated treatment groups. (**H, I**) Neutrophil extracellular trap (NET) formation detected by fluorescence microscopy using Helix dye in human neutrophils activated with platelets. Cells were unstimulated (no chemokine, NC), stimulated with CCL28 alone, or with CCL28 and the CCR3 agonist SB328737 and/or the CCR10 agonist BI-6901, as indicated. (**H**) Representative images of fluorescence microscopy with DAPI (blue) and Helix (green). (**I**) Quantification of NETs represented as percentage of cells forming NETs based on observed morphology. Connected circles represent NET abundance in cell populations from the same donor following different indicated treatments. (**A–E**) Bars represent the mean ± SD. (**A–C**) Data were analyzed by non-parametric analysis of variance (ANOVA) (Kruskal–Wallis's test), assuming non-equal SD given the differences in the variance among the groups, followed by Dunn's multiple comparisons test. (**D, I**) Data were analyzed by ratio paired *t* test. (**E–G**) Log-transformed data were analyzed by one-way ANOVA for paired samples. Greenhouse–Geisser correction was applied in **F** and **G** given the differences in variance among the groups. Tukey's multiple comparison test was performed to compare all conditions to each other. (**I**) Ratio paired *t* tests were used to compare NET levels in samples from the same donor. Significant changes are indicated by *p ≤ 0.05, **p ≤ 0.01, ***p ≤ 0.001, ****p ≤ 0.0001; ns, not significant.

The online version of this article includes the following figure supplement(s) for figure 5:

**Figure supplement 1.** Neutrophil extracellular trap (NET) formation (Helix$^+$ MPO$^+$ neutrophils) detected by flow cytometry in human neutrophils activated with platelets.

dyes DAPI and HELIX, and evaluated NET formation by fluorescence microscopy (*Figure 5H*). Incubation with activated platelets and CCL28 increased the percentage of NETs compared to neutrophils not stimulated with CCL28 (*Figure 5H, I*). Complementary experiment, analyzing DNA–MPO complexes confirmed an increased percentage of DNA–MPO complexes in response to platelet and CCL28 stimulation (*Figure 5—figure supplement 1*). The effect of CCL28 on platelet-activated NET formation was primarily mediated by CCR3, as the CCR3 antagonist SB328437 significantly reduced the percentage of observable NET$^+$ neutrophils (*Figure 5H1*) and DNA–MPO complexes (*Figure 5—figure supplement 1*). In contrast, the CCR10 antagonist BI-6901 did not significantly reduce NET formation, and combined antagonism of CCR3 and CCR10 had an effect similar to CCR3 antagonism alone (*Figure 5H1*, *Figure 5—figure supplement 1*). Together, these results demonstrate that CCL28 enhances neutrophil ROS production and NET formation primarily in a CCR3-dependent manner.

## Discussion

The mucosal immune response serves to maintain tissue homeostasis and to protect the host against invading pathogens. Here, we discovered that the chemokine CCL28 significantly contributes to neutrophil accumulation and activation in the mucosa during gastrointestinal infection with *Salmonella* and lung infection with *Acinetobacter*.

Consistent with our initial observation that *Ccl28*$^{−/−}$ mice exhibit higher mortality during STm infection (*Burkhardt et al., 2019*), we found higher intestinal colonization and extraintestinal dissemination of STm in *Ccl28*$^{−/−}$ mice compared to their wild-type littermates (*Figure 1*). This beneficial role for CCL28 was negligible when the pathogen was inoculated intraperitoneally to bypass the gut mucosa (*Figure 1—figure supplement 2*). Although CCL28 exerts direct antimicrobial activity against some bacteria and fungi (*Hieshima et al., 2003*), it does not directly inhibit STm wild-type in vitro (*Figure 1—figure supplement 2*). Although CCL28 receptors CCR3 and CCR10 are expressed on eosinophils and on B and T cells (*Pan et al., 2000*; *Höchstetter et al., 2000*; *Wang et al., 2000*), the protective role of CCL28 during *Salmonella* infection does not seem to involve these cell types, as they did not vary in abundance between wild-type and *Ccl28*$^{−/−}$ mice during infection (*Figure 1—figure supplements 4 and 5*). However, it is still possible that CCL28 modulates B and T cell responses in chronic model of *Salmonella* infection, which could be explored in future studies using attenuated *Salmonella* strains

(*Hapfelmeier et al., 2005*), or mice genetically more resistant to *Salmonella* because they express a functional Nramp1 (*Monack et al., 2004*).

Neutrophils are a hallmark of inflammatory diarrhea and are rapidly recruited to the gut following infection in the *Salmonella* colitis model. We found that neutrophil numbers were significantly reduced in the mucosa of infected *Ccl28*−/− relative to wild-type mice (*Figure 1*), identifying CCL28 as a key factor for neutrophil accumulation during infection. Neutrophils migrate from the bone marrow to the blood and to infected sites following a chemokine gradient (*Capucetti et al., 2020*), expressing various chemokine receptors including CXCR1, CXCR2, CXCR4, and CCR2, and under certain circumstances, CCR1 and CCR6 (*Kobayashi, 2008*). CXCR2 is a promiscuous receptor that binds to the chemokines CXCL1, 2, 3, 5, 6, 7, and 8 (*Ahuja and Murphy, 1996*), whereas CXCR1 only binds CXCL6 and CXCL8 (*Capucetti et al., 2020*). Activation of CXCR2 mobilizes neutrophils from the bone marrow to the bloodstream and participates in NET release (*Marcos et al., 2010*). Engagement of CXCR1 and CXCR2 also triggers signaling pathways boosting production of cytokines and chemokines that amplify neutrophil responses (*Sabroe et al., 2005*). Following extravasation to the site of infection, neutrophils downregulate CXCR2 and upregulate CCR1, 2, and 5, which cumulatively boosts neutrophil ROS production and phagocytic activity (*Capucetti et al., 2020*). Our results indicate that CCL28 contributes to neutrophil accumulation and activation (*Figure 1*), with its receptors CCR3 and CCR10 upregulated in the mucosa during infection, where up to ~50% of neutrophils express surface CCR3 (*Figure 3*). The reciprocal regulation of CXCR2 and CCR3/CCR10 in neutrophils and each receptor's contribution to neutrophil migration and retention during infectious colitis requires further study.

Although an initial study concluded CCR3 was absent on neutrophils (*Höchstetter et al., 2000*), subsequent studies reported CCR3 expression on human neutrophils isolated from patients with chronic lung disease (*Hartl et al., 2008*) and on neutrophils isolated from the BAL fluid of mice infected with influenza (*Rudd et al., 2019*). Our study demonstrates that a substantial number of neutrophils isolated from infected mucosal sites express CCR3, and fewer express CCR10 on their surface, while resting neutrophils do not express these receptors on their surface (*Figure 3*). The rapid surface expression of CCR3 on neutrophils upon infection suggests that the receptor is stored intracellularly, similar to eosinophils (*Spencer et al., 2006*). Indeed, neutrophils isolated from bone marrow, blood, and infected mucosal tissue were all positive for CCR3 intracellular staining (*Figure 4*). In vitro, we could recapitulate the increase in surface receptor expression by incubating bone marrow neutrophils with proinflammatory stimuli (LPS, or the cytokines GM-CSF + IFNγ + TNFα) or following phagocytosis of bacterial pathogens (*Figure 3*). CCL28 stimulation of bone marrow neutrophils in vitro increased their antimicrobial activity and ROS production during *Salmonella* infection, which was reverted by blocking CCR3 but not CCR10 (*Figure 5*). Platelet-activated neutrophils stimulated with CCL28 also showed enhanced NET formation, largely in a CCR3-dependent manner (*Figure 5*). Thus, CCL28 is a potent activator of neutrophils, primarily via CCR3. Further studies with receptor knockout mice are needed to determine the contribution of each CCL28 receptor to the in vivo phenotypes.

A reduction of neutrophil accumulation was also observed in the BAL and lung of *Ccl28*−/− mice during *Acinetobacter* infection (*Figure 2*), with neutrophils recruited to the lung harboring surface CCR3 and CCR10 (*Figures 3 and 4*). However, the functional consequence of CCL28 deficiency was strikingly different in this model, as *Ccl28*−/− mice were protected during Ab pneumonia. Most *Ccl28*−/− mice survived until the experiment's endpoint at 10 dpi, whereas nearly all wild-type littermates succumbed by 2 dpi (*Figure 2*). The lung, possessing a thin, single-cell alveolar layer to promote gas exchange, is less resilient than the intestine to neutrophil inflammation before losing barrier integrity and critical functions. Thus, although insufficient neutrophil recruitment can lead to life-threatening infection, extreme accumulation of neutrophils can result in excessive inflammatory lung injury (*Craig et al., 2009*). The high survival of *Ccl28*−/− mice infected with Ab indicates that CCL28 may be detrimental for the host in the context of some pulmonary infections. While functioning neutrophils have been described to play a role in controlling *Acinetobacter* infection (*van Faassen et al., 2007*; *García-Patiño et al., 2017*; *Grguric-Smith et al., 2015*), excessive neutrophil recruitment can exacerbate lung injury (*Yamada et al., 2013*; *Zeng et al., 2020*; *Zeng et al., 2019*). For instance, depletion of alveolar macrophages in one *Acinetobacter* pneumonia study increased neutrophil infiltration, promoted tissue damage, and increased systemic dissemination of the pathogen (*Lee et al., 2020*). In contrast to *Salmonella*, CCL28 stimulation did not enhance neutrophil antimicrobial activity against *Acinetobacter*, which may partly explain the lack of a protective response (*Figure 5*). Further investigation is

required to understand why *Acinetobacter* may be resistant to CCL28-dependent neutrophil antimicrobial responses.

Even though CCL28 exhibited direct antimicrobial activity against *Acinetobacter*, higher concentrations of CCL28 (1 µm) are needed for protection and were not sufficient against higher pathogen burdens (*Figure 1—figure supplement 2*). These findings align with prior studies indicating that the host response to infection can be context-dependent, with some immune components mediating different outcomes in the gut and in the lung. For example, *Cxcr2*$^{-/-}$ mice exhibit a defect in neutrophil recruitment that is detrimental during *Salmonella* infection (*Marchelletta et al., 2015*) but protective during lung infection with *Mycobacterium tuberculosis* due to reduced neutrophil recruitment and reduced pulmonary inflammation (*Nouailles et al., 2014*). Similarly, CCL28-dependent modulation of neutrophil accumulation and activation during infection can be protective or detrimental depending on the pathogen and the site of infection.

Overall, this study demonstrates that CCL28 plays an important role in the mucosal response to pathogens by promoting neutrophil accumulation at the site of infection. Neutrophils isolated from infected mucosa express the CCL28 receptors CCR3 and CCR10, and CCL28 enhances neutrophil activation, ROS production, and NET formation, primarily through CCR3. These findings have implications for other infectious and non-infectious diseases where neutrophil recruitment plays a major role, and may lead to the identification of CCL28-targeted therapies to modulate neutrophil function and mitigate collateral damage.

# Materials and methods

## Key resources table

| Reagent type (species) or resource | Designation | Source or reference | Identifiers | Additional information |
|---|---|---|---|---|
| Strain, strain background (*Salmonella enterica*) | *S. enterica* serovar Typhimurium strain IR715 | Lab stock; PMID:7868611 | | Nalidixic acid-resistant derivative of strain ATCC 14028s |
| Strain, strain background (*Salmonella enterica*) | *S. Typhimurium* IR715 *ΔphoQ* | Lab stock; from Michael McClelland PMID:19578432 | | PhoQ coding sequence disrupted by a kanamycin cassette |
| Strain, strain background (*Escherichia coli*) | *E. coli* K12 strain MG1655 | Lab Stock | ATCC Cat#700926 | |
| Strain, strain background (*Acinetobacter baumannii*) | *A. baumannii* strain AB5075 | Walter Reed Medical Center; PMID:24865555 | | |
| Genetic reagent (*Mus musculus*) | C57BL/6 *Ccl28*::*Neo*$^r$ | Deltagen; PMID:30855201 | | Obtained from Albert Zlotnik (UC Irvine); Allelic exchange into Ccl28 |
| Genetic reagent (*Mus musculus*) | C57BL/6 *Ccl28*$^{-/-}$ (C57BL/6JCya-*Ccl28*$^{em1}$/Cya) | Cyagen Biosciences | Product Number: S-KO-17095; RRID:MGI:1861731 | Generated by CRISPR/Cas9-mediated deletion of exons 1–3 |
| Biological sample (*Homo sapiens*) | Primary human blood neutrophils | Human volunteers, UNAM | | Freshly isolated from human volunteers |
| Biological sample (*Mus musculus*) | Primary bone marrow cells | C57BL/6 *Ccl28*$^{+/+}$ mice, UC San Diego | | Freshly isolated from wild-type mice of the *Ccl28* colony |
| Antibody | Anti-mouse CD16/CD32 (Rat monoclonal; unconjugated Fc Block) | BioLegend | Clone: 93; Cat#101302; RRID:AB_312801 | FC (1:50) |
| Antibody | Anti-mouse CD45 (Rat monoclonal; Pacific Blue) | BioLegend | Clone: 30-F11; Cat#103126; RRID:AB_493535 | Sony SA3800 FC (1:800); FACSCantoII FC (1:400) |
| Antibody | Anti-mouse/human CD11b (Rat monoclonal; Spark Blue 550) | BioLegend | Clone: M1/70; Cat#101290; RRID:AB_2922452 | FC (1:400) |
| Antibody | Anti-mouse Ly6G (Rat monoclonal; Brilliant Violet 421) | BioLegend | Clone: 1A8; Cat#127628; RRID:AB_2562567 | FC (1:1600) |
| Antibody | Anti-mouse CD170 (SiglecF) (Rat monoclonal; PE/Dazzle 594) | BioLegend | Clone: S17007L; Cat#155530; RRID:AB_2890716 | FC (1:400) |

*Continued on next page*

*Continued*

| Reagent type (species) or resource | Designation | Source or reference | Identifiers | Additional information |
|---|---|---|---|---|
| Antibody | Anti-mouse CCR3 (Rat monoclonal; PE) | R&D Biosystems | Clone: 83103; Cat#FAB729P; RRID:AB_2074151 | FC (1:100) |
| Antibody | Anti-mouse CCR10 (Rat monoclonal; APC) | R&D Biosystems | Clone: 248918; Cat#FAB2815; RRID:AB_1151964 | FC (1:100) |
| Antibody | Anti-mouse CD11c (Armenian Hamster monoclonal; Brilliant Violet 421) | BioLegend | Clone: N418; Cat#117343; RRID:AB_2563099 | FC (1:400) |
| Antibody | Anti-mouse Ly6G (Rat monoclonal; FITC) | BioLegend | Clone: 1A8; Cat#127606; RRID:AB_1236494 | FC (1:400) |
| Antibody | Anti-mouse CD170 (SiglecF) (Rat monoclonal; FITC) | BioLegend | Clone: S17007L; Cat#155503; RRID:AB_2750232 | FC (1:400) |
| Antibody | Anti-mouse F4/80 (Rat monoclonal; PE/Dazzle 594) | BioLegend | Clone: BM8; Cat#123146; RRID:AB_2564133 | FC (1:400) |
| Antibody | Anti-mouse CD8a (Rat monoclonal; Brilliant Violet 421) | BioLegend | Clone: 53-6.7; Cat#100737; RRID:AB_10897101 | FC (1:1600) |
| Antibody | Anti-mouse CD3 (Rat monoclonal; FITC) | BioLegend | Clone: 17A2; Cat#100204; RRID:AB_312661 | FC (1:400) |
| Antibody | Anti-mouse CD4 (Rat monoclonal; PerCP/Cyanine5.5) | BioLegend | Clone: RM4-5; Cat#100539; RRID:AB_893332 | FC (1:800) |
| Antibody | Anti-mouse CD8a (Rat monoclonal; PE) | BioLegend | Clone: 53-6.7; Cat#100708; RRID:AB_312747 | FC (1:1600) |
| Antibody | Anti-mouse CD19 (Rat monoclonal; Alexa Fluor 700) | BioLegend | Clone: 6D5; Cat#115528; RRID:AB_493735 | FC (1:400) |
| Antibody | Anti-mouse/human CD11b (Rat monoclonal; APC) | BioLegend | Clone: M1/70; Cat#101212; RRID:AB_312795 | FC (1:800) |
| Antibody | Anti-mouse/human CD11b (Rat monoclonal; Brilliant Violet 510) | BioLegend | Clone: M1/70; Cat#101245; RRID:AB_2561390 | FC (1:400) |
| Antibody | Anti-mouse F4/80 (Rat monoclonal; FITC) | BioLegend | Clone: BM8; Cat#123108; RRID:AB_893502 | FC (1:200) |
| Antibody | Anti-mouse Ly6G (Rat monoclonal; PerCP) | BioLegend | Clone: 1A8; Cat#127654; RRID:AB_2616999 | FC (1:400) |
| Antibody | Anti-mouse CD170 (SiglecF) (Rat monoclonal; APC) | BioLegend | Clone: S17007L; Cat#155508; RRID:AB_2750237 | FC (1:400) |
| Antibody | Anti-mouse CD11c (Armenian Hamster monoclonal; PE/Cyanine7) | BioLegend | Clone: N418; Cat#117317; RRID:AB_493569 | FC (1:400) |
| Antibody | Anti-mouse CD19 (Rat monoclonal; PE/Cyanine7) | BioLegend | Clone: 6D5; Cat#115520; RRID:AB_313655 | FC (1:400) |
| Antibody | Anti-mouse CCR3 (Rat monoclonal; unconjugated) | R&D Systems | Clone: 83103; Cat#MAB1551; RRID:AB_2074150 | In vitro signaling blockade (5 µg/100 µl) |
| Antibody | Anti-mouse CCR10 (Rat monoclonal; unconjugated) | R&D Systems | Clone: 248918; Cat#MAB2815; RRID:AB_2074258 | In vitro signaling blockade (5 µg/100 µl) |
| Antibody | Rat IgG2A Isotype Control Antibody (Rat monoclonal; unconjugated) | R&D Systems | Clone: 54447; Cat#MAB006; RRID:AB_357349 | In vitro signaling blockade (5 µg/100 µl) |
| Antibody | Anti-mouse Ly6G (Rat monoclonal; unconjugated) | BioLegend | Clone: 1A8; Cat#127601; RRID:AB_1089179 | Lung neutrophil IF (1:100) |
| Antibody | Goat Anti-rat IgG (H+L) Cross-Adsorbed Secondary Antibody (Goat polyclonal; Alexa Fluor 555) | Invitrogen | Cat#A-21434; RRID:AB_2535855 | Lung neutrophil IF: (1:400) |
| Antibody | Human TruStain FcX (Human monoclonal mix; unconjugated Fc Receptor blocking solution) | BioLegend | Cat#422302; RRID:AB_2818986 | FC (1:100) |

*Continued on next page*

*Continued*

| Reagent type (species) or resource | Designation | Source or reference | Identifiers | Additional information |
|---|---|---|---|---|
| Antibody | Anti-human CD45 (Mouse monoclonal; PerCP/Cyanine5.5) | BioLegend | Clone: HI30; Cat#304028; RRID:AB_893338 | FC (1:300) |
| Antibody | Anti-mouse/human CD11b (Rat monoclonal; Pacific Blue) | BioLegend | Clone: M1/70; Cat#101224; RRID:AB_755986 | FC (1:200) |
| Antibody | Anti-human CD62L (Mouse monoclonal; FITC) | BioLegend | Clone: DREG-56; Cat#304838; RRID:AB_2564162 | FC (1:300) |
| Antibody | Anti-human CCR3 (Rat monoclonal; PE) | R&D Systems | Clone: 61828; Cat#FAB155P; RRID:AB_2074157 | FC (1:100) |
| Antibody | Anti-human CCR10 (Rat monoclonal; APC) | R&D Systems | Clone: 314305; Cat#FAB3478A; RRID:AB_573043 | FC (1:100) |
| Antibody | Anti-human myeloperoxidase (Mouse monoclonal; Biotin-conjugated) | Novus Biologicals | Clone MPO421-8B2; Cat#NBP2-41406B | FC (1:50) |
| Sequence-based reagent | Mouse *Actb* qPCR primers | IDT | Forward: GGCTGTATTCCCCTCCATCG; Reverse: CCAGTTGGTAACAATGCCATGT | |
| Sequence-based reagent | Mouse *Cxcl1* qPCR primers | IDT | Forward: TGCACCCAAACCGAAGTCAT; Reverse: TTGTCAGAAGCCAGCGTTCAC | |
| Sequence-based reagent | Mouse *Tnf* qPCR primers | IDT | Forward: CATCTTCTCAAAATTCGAGTGACAA; Reverse: TGGGAGTAGACAAGGTACAACCC | |
| Sequence-based reagent | Mouse *Ifng* qPCR primers | IDT | Forward: TCAAGTGGCATAGATGTGGAAGAA; Reverse: TGGCTCTGCAGGATTTTCATG | |
| Sequence-based reagent | Mouse *Csf3* qPCR primers | IDT | Forward: TGCTTAAGTCCCTGGAGCAA; Reverse: AGCTTGTAGGTGGCACACAA | |
| Sequence-based reagent | Mouse *Il1b* qPCR primers | IDT | Forward: CTCTCCAGCCAAGCTTCCTTGTGC; Reverse: GCTCTCATCAGGACAGCCCAGGT | |
| Sequence-based reagent | Mouse *Il17a* qPCR primers | IDT | Forward: GCTCCAGAAGGCCCTCAGA; Reverse: AGCTTTCCCTCCGCATTGA | |
| Peptide, recombinant protein | Recombinant Mouse CCL28 (MEC) | BioLegend | Cat#584706 | In vitro killing: various concentrations (indicated in text) |
| Peptide, recombinant protein | Recombinant Mouse CCL28 Protein | R&D Systems | Cat#533-VI | Chemotaxis: 50 nM; neutrophil stimulation: 50 nM |
| Peptide, recombinant protein | Recombinant Mouse CCL11/Eotaxin Protein | R&D Systems | Cat#420-ME | Chemotaxis: 50 nM; neutrophil stimulation: 25 nM |
| Peptide, recombinant protein | Recombinant Murine KC (CXCL1) | Peprotech | Cat#250–11 | Chemotaxis: 50 nM |
| Peptide, recombinant protein | Recombinant human CCL28 | BioLegend | Cat#584602 | Neutrophil stimulation: 50 nM |
| Peptide, recombinant protein | Recombinant Mouse TNF-α | BioLegend | Cat#575202 | Neutrophil stimulation: 100 ng/ml |
| Peptide, recombinant protein | Recombinant Mouse IFN-γ | BioLegend | Cat#575304 | Neutrophil stimulation: 500 U/ml |

*Continued on next page*

*Continued*

| Reagent type (species) or resource | Designation | Source or reference | Identifiers | Additional information |
|---|---|---|---|---|
| Peptide, recombinant protein | Recombinant Mouse GM-CSF | BioLegend | Cat#576302 | Neutrophil stimulation: 10 ng/ml |
| Peptide, recombinant protein | LPS-B5 Ultrapure | Invivogen | Cat#tlrl-pb5lps | Mouse neutrophil stimulation: 100 ng/ml |
| Commercial assay or kit | EasySep Mouse Neutrophil Enrichment Kit | STEMCELL Technologies | Cat#19762 | |
| Commercial assay or kit | EasySep Direct Human Neutrophil Isolation Kit | STEMCELL Technologies | Cat#19666 | |
| Commercial assay or kit | Mouse CCL28 ELISA Max Deluxe | BioLegend | Cat# 441304 | |
| Commercial assay or kit | Mouse Myeloperoxidase DuoSet ELISA Kit | R&D Systems | Cat#DY3667 | |
| Commercial assay or kit | Mouse Neutrophil Elastase/ELA2 DuoSet ELISA Kit | R&D Systems | Cat#DY4517 | |
| Commercial assay or kit | Mouse S100a9 DuoSet ELISA Kit | R&D Systems | Cat#DY2065 | |
| Commercial assay or kit | PowerUp SYBR Green Master Mix for qPCR | Applied Biosystems (Thermo Fisher) | Cat#A25742 | |
| Commercial assay or kit | SuperScript VILO cDNA Synthesis Kit | Thermo Fisher | Cat#11766500 | |
| Commercial assay or kit | eBioscience Fixable Viability Dye eFluor 780 | Thermo Fisher | Cat#65-0865-14 | FC (1:1000) |
| Chemical compound, drug | fMLP (N-Formyl-Met-Leu-Phe) | Sigma-Aldrich | Cat#F3506 | Neutrophil stimulation: 1 µM |
| Chemical compound, drug | PMA (Phorbol 12-myristate 13-acetate) | Sigma-Aldrich | Cat#79346 | Neutrophil stimulation: 100 nM |
| Chemical compound, drug | Cytochalasin D | Sigma-Aldrich | Cat#C8273 | Incubated cells at 10 µM |
| Chemical compound, drug | SB328437 [$N$-(1-naphthalenylcarbonyl)-4-nitro-L-phenylalanine methyl ester] | Tocris Bioscience | Cat#3650 | CCR3 antagonist (10 µM) |
| Chemical compound, drug | BI-6901 ($N$-[(1$R$)-3-(2-Cyano-1$H$-pyrrol-1-yl)-1-[(4-methyl-1-piperidinyl)carbonyl]propyl]-1$H$-indole-4-sulfonamide) | Gift from Boehringer-Ingelheim Pharma GmbH & Co KG | | CCR10 antagonist (20 µM) |
| Chemical compound, drug | Xylazine | VetOne | Cat#RX-0065 | Used for temporary anesthesia: 10 mg/kg, i.p. |
| Chemical compound, drug | Ketamine | Zoetis | Cat#000680 | Used for temporary anesthesia: 100 mg/kg, i.p |
| Chemical compound, drug | Nalidixic acid sodium salt | Fisher Scientific | Cat#AAJ6355014 | 50 µg/ml for selection |
| Chemical compound, drug | Streptomycin sulfate | Fisher Scientific | Cat#5711 | For oral gavage (20 mg/mouse) |
| Software, algorithm | GraphPad Prism 10.0 | GraphPad Software | RRID:SCR_002798 | |
| Software, algorithm | FlowJo 10.8.1 | BD Biosciences | RRID:SCR_008520 | |
| Software, algorithm | QuantStudio 5 Reat-Time PCR System | Thermo Fisher Scientific | RRID:SCR_020240 | |
| Software, algorithm | QuPath Analysis Software | QuPath (PMID:29203879) | RRID:SCR_018257 | |
| Other | DMSO | Millipore Sigma | Cat#EM-MX1458-6 | Used at 0.1% for vehicle for cytochalasin D during in vitro infection assays described in the Materials and methods |

*Continued on next page*

*Continued*

| Reagent type (species) or resource | Designation | Source or reference | Identifiers | Additional information |
|---|---|---|---|---|
| Other | 2',7'-Dichlorodihydrofluorescein diacetate | Invitrogen | Cat#D399 | Used at 25 µM for incubation of neutrophils for detection of ROS production by neutrophils, as described in the Materials and methods |
| Other | TRI Reagent | Sigma-Aldrich | Cat#T9424 | Used for RNA isolation from tissues, described in Materials and methods section 'RNA extraction and qPCR' |
| Other | SlowFade Gold Antifade Mountant | Invitrogen | Cat#36936 | Used for staining and mounting immunoflourescent lung sections, described in Materials and methods section 'Immunofluorescence' |
| Other | APC/Cy7 Streptavidin | BioLegend | Cat#405208 | For tagging biotin-conjugated anti-human myeloperoxidase; FC (1:1000) |
| Other | OneComp eBeads | Thermo Fisher | Cat#01-1111-42 | Added to cells at $5 \times 10^5$ beads per $1 \times 10^6$ cells, as described in the Materials and methods section 'In vitro neutrophil stimulation' |
| Other | Collagenase, Type VIII | Sigma-Aldrich | Cat#C2139 | For tissue digestion, as described in the Materials and methods: 1 mg/ml |
| Other | Liberase | Sigma-Aldrich | Cat#5401020001 | For tissue digestion, as described in the Materials and methods: 20 µg/ml |
| Other | DNase I | Sigma-Aldrich | Cat#DN25 | For tissue digestion, as described in the Materials and methods: 0.25 mg/ml |
| Other | Helix NP Green | BioLegend | Cat#425303 | For staining neutrophil DNA, as described in the Materials and methods. FC: 10 nM; immuno-fluorescence: 5 µM |
| Other | LB Broth, Miller | Fisher Scientific | Cat#DF0446-17-3 | Used for routine culturing of *S.* Typhimurium, described in Materials and methods section '*Salmonella* infection models' |
| Other | LB agar, Miller | Fisher Scientific | Cat#DF0445-17-4 | Used for growth and enumeration of *S.* Typhimurium and *Acinetobacter* CFUs, as described throughout the Materials and methods section |
| Other | Mueller-Hinton Broth | Fisher Scientific | Cat#DF0757-17-6 | Used for routine culturing of *A. baumannii*, described in Materials and methods section '*Acinetobacter* infection model' |
| Other | DPBS | Gibco | Cat#14190250 | Used for washing or resuspension of various cells and bacteria, as described throughout the Materials and methods section |
| Other | cOmplete, Mini, EDTA-free Protease Inhibitor Cocktail | Sigma-Aldrich | Cat#4693159001 | Used for fecal protease inhibition as described in the Materials and methods |
| Other | Fetal bovine serum (FBS), heat-inactivated | Gibco | Cat#A3840001 | Used for general cell preservation and assays as described in the Materials and methods |
| Other | Antibiotic–antimycotic | Gibco | Cat#15-240-062 | Used for general tissue cell preservation as described in the Materials and methods |
| Other | RPMI 1640 Medium, with L-glutamine | Gibco | Cat#11875-119 | Used for general tissue cell preservation and assays as described in the Materials and methods |
| Other | RPMI 1640 Medium, no glutamine, no phenol red | Gibco | Cat#32404014 | Used for $H_2$DCFDA ROS assays as described in the Materials and methods |

*Continued on next page*

*Continued*

| Reagent type (species) or resource | Designation | Source or reference | Identifiers | Additional information |
|---|---|---|---|---|
| Other | IMDM | Gibco | Cat#12440061 | Used for gut tissue cell isolation as described in the Materials and methods |
| Other | Hank's Balanced Salt Solution | Fisher Scientific | Cat#MT21021CV | Used for gut tissue cell isolation as described in the Materials and methods |
| Other | HEPES | Gibco | Cat#15630080 | Used for general tissue cell preservation and assays as described in the Materials and methods |
| Other | EDTA | Fisher Scientific | Cat#S311-500 | Used for collection of mouse blood, and for lung and gut tissue cells isolation as described in Materials and methods section 'Cell extraction and analysis' |
| Other | Bovine serum albumin (BSA) | Fisher Scientific | Cat#BP9703100 | Added to various media for the purpose of blocking non-specific interactions, as described in the Materials and methods sections 'Cell extraction and analysis' and 'Chemotaxis assay' |

## Generation and breeding of *Ccl28*$^{-/-}$ mice

The first colony of *Ccl28*$^{-/-}$ mice was described in a prior manuscript (**Burkhardt et al., 2019**) and used for initial studies at UC Irvine. At UC San Diego, we generated a new colony of *Ccl28*$^{-/-}$ mice with Cyagen Biosciences (Santa Clara, California), using CRISPR/CAS9 technology. Exons 1 and 3 were selected as target sites, and two pairs of gRNA targeting vectors were constructed and confirmed by sequencing. The gRNA and Cas9 mRNA were generated by in vitro transcription, then co-injected into fertilized eggs for knockout mouse production. The resulting pups (F0 founders) were genotyped by PCR and confirmed by sequencing. F0 founders were bred to wild-type mice to test germline transmission and for F1 animal generation. Tail genotyping of offspring was performed using the following primers:

F: 5'-TCATATACAGCACCTCACTCTTGCCC-3', R: 5'-GCCTCTCAAAGTCATGCCAGAGTC-3' and He/Wt-R: 5'-AGGGTGTGAGGTGTCCTTGATGC -3'. The expected product size for the wild-type allele is 466 bp and for the knockout allele is 700 bp.

All mouse experiments were conducted with cohoused wild-type and *Ccl28*$^{-/-}$ littermate mice, and were reviewed and approved by the Institutional Animal Care and Use Committees at UC Irvine (protocol #2009-2885) and UC San Diego (protocols #S17107 and #S00227M).

## *Salmonella* infection models

For the *Salmonella* colitis model, 8- to 10-week-old male and female mice were orally gavaged with 20 mg streptomycin 24 hr prior to oral gavage with $10^9$ CFU of *S. enterica* serovar Typhimurium strain IR715 (a fully virulent, nalidixic acid-resistant derivative of ATCC 14028s) (**Stojiljkovic et al., 1995**), as previously described (**Barthel et al., 2003**; **Walker et al., 2023**; **Raffatellu et al., 2009**). Mice were euthanized at 2 or 3 days post-infection, then colon content, spleen, mesenteric lymph nodes, Peyer's patches, blood, and bone marrow were collected, weighed, homogenized, serially diluted, and plated on Miller Lysogeny broth (LB) + Nal (nalidixic acid, 50 μg/ml) agar plates to enumerate *Salmonella* CFU. Mice displaying extremely poor colonization in 1 dpi ($\leq 10^3$ CFU/mg feces) or extremely high weight loss 1 dpi ($\geq 8\%$ loss from the day of infection) were excluded from downstream analyses due to likely technical errors during inoculation. For the *Salmonella* bacteremia model, mice were injected intraperitoneally with $10^3$ CFU. Mice were euthanized at 4 days post-infection, then blood, spleen, and liver were collected to determine bacterial counts.

## *Acinetobacter* infection model

For the murine pneumonia model**,** *A. baumannii* strain AB5075 was cultured in Cation-Adjusted Mueller-Hinton Broth (CA-MHB) overnight, then subcultured the next day to an OD$_{600}$ of ~0.4 ($1 \times 10^8$ CFU/ml; mid-logarithmic phase). These cultures were centrifuged at $3202 \times g$ for 10 min, the supernatant was removed, and pellets were resuspended and washed in an equal volume of 1× Dulbecco's

PBS (DPBS) three times. The final pellet was resuspended in 1× DPBS to yield a suspension of 2.5 × $10^9$ CFU/ml. Using an operating otoscope (Welch Allyn), mice under 100 mg/kg ketamine (Koetis) + 10 mg/kg xylazine (VetOne) anesthesia were inoculated intratracheally with 40 µl of the bacterial suspension ($10^8$ CFU/mouse). Post-infection mice were recovered on a sloped heating pad. For analysis of bacterial CFU, mice were sacrificed 1 day post-infection, the BAL, blood, and lungs were collected, and serial dilutions were plated on LB agar to enumerate bacteria (*Dillon et al., 2019*).

## CCL28 ELISA

Fresh fecal and blood samples were collected at 4 days post-infection from wild-type mice for quantification of CCL28. Fecal pellets were weighed, resuspended in 1 ml of sterile PBS containing a protease inhibitor cocktail (Roche), and incubated at room temperature shaking for 30 min. Whole-blood samples were collected by cardiac puncture and allowed to clot at room temperature for 30 min. Samples were centrifuged at 9391 × *g* for 10 min, supernatant/serum was collected, then analyzed to quantify CCL28 using a sandwich ELISA kit (BioLegend).

## Cell extraction and analysis

For the *Salmonella* colitis model, the terminal ileum, cecum, and colon were collected at indicated time points, either 2 or 3 days post-infection. All tissues were kept in Iscove's Modified Dulbecco's Medium (IMDM) supplemented with 10% fetal bovine serum (FBS, Gibco) and 1% antibiotic/antimycotic (Gibco). Next, any Peyer's patches were removed, and the intestinal fragments were cut open longitudinally and washed with Hank's Balanced Salt Solution (HBSS) supplemented with 15 mM 4-(2-hydroxyethyl)-1-piperazineethanesulfonic acid (HEPES) and 1% antibiotic/antimycotic. Then, the tissue was shaken in 10 ml of an HBSS/15 mM HEPES/5 mM ethylenediaminetetraacetic acid (EDTA)/10% FBS solution at 37°C for 15 min. The supernatant was removed and kept on ice. The remaining tissue was cut into small pieces and digested in a 10 ml mixture of collagenase (Type VIII, 1 mg/ml), Liberase (20 µg/ml), and DNAse (0.25 mg/ml) in IMDM medium for 15 min, shaking at 37°C. Afterwards, the supernatant and tissue fractions were strained through a 70-µm cell strainer and pooled, and the extracted cells were used for flow cytometry staining. For the *A. baumannii* infection model, the lungs were collected, minced, and processed with collagenase and DNase as described above for the gut. BAL was collected by instilling 1 ml DPBS/10 mM EDTA via the trachea into the lungs, and recovering the majority (~700–900 µl) into a syringe after 20 s. The lavage fluid was centrifuged, and pellets were washed with 1× PBS. Samples where less than 500 µl of the fluid was recovered (indicating improper syringe placement during collection) were excluded from downstream analyses. The obtained cells were used for flow cytometry staining. Briefly, cells were blocked with a CD16/32 antibody (BioLegend), stained with the fixable viability dye eFluor780 (Thermo Fisher), then extracellularly stained using the following conjugated monoclonal antibodies: anti-mouse CD45 (clone 30-F11), anti-mouse CD3 (clone 17A2), anti-mouse CD4 (clone RM4-5), anti-mouse CD8α (clone 53-6.7), anti-mouse CD19 (clone 1D3/CD19), anti-mouse Ly6G (clone 1A8), anti-mouse CD11b (clone M1/70), anti-mouse SiglecF (clone S17007L), anti-mouse F4/80 (clone BM8), anti-mouse CD11c (clone N418) from BioLegend; anti-mouse CCR3 (clone 83101), and anti-mouse CCR10 (clone 248918) from R&D Systems. After staining, cells were washed with DPBS + 0.5% bovine serum albumin (BSA) and either immediately analyzed on a SA3800 flow cytometer (Sony Biotechnology), or first fixed for 20 min with 4% paraformaldehyde (Fixation buffer; BioLegend) and analyzed later. When intracellular staining was performed, cells were permeabilized in Permeabilization buffer (BioLegend), re-blocked with the CD16/32 antibody, and the staining was performed in the same buffer following the manufacturer's instructions. In different experiments, cells were analyzed using a SA3800 Spectral Cell analyzer, a BD FACSCanto II flow cytometer (BD Biosciences), and a LSRII flow cytometer (BD Biosciences), and the collected data were analyzed with FlowJo v10 software (TreeStar). For analysis of human neutrophils, whole-blood samples were collected in EDTA for cellular analyses. Whole-blood cell staining was performed using an Fc receptor blocking solution (Human TruStain FcX; BioLegend), the viability dye eFluor780 (Thermo Fisher), and the following conjugated monoclonal antibodies: PerCP/Cy5.5 anti-human CD45 antibody (clone HI30), Pacific Blue anti-mouse/human CD11b antibody (clone M1/70), FITC anti-human CD62L antibody (clone DREG-56), from BioLegend; PE anti-human CCR3 antibody (clone 61828),

and APC anti-human CCR10 antibody (clone 314305) from R&D Systems. Samples were analyzed by flow cytometry using an LSR Fortessa flow cytometer (BD Biosciences), and data were analyzed using FlowJo v10 software.

## In vitro neutrophil stimulation

Neutrophils were obtained from the bone marrow of C57BL/6 wild-type mice using the EasySep Mouse Neutrophil Enrichment Kit (STEMCELL), following the manufacturer's instructions. After enrichment, $1 \times 10^6$ neutrophils were seeded per well in a round-bottom 96-well plate with Roswell Park Memorial Institute (RPMI) media supplemented with 10% FBS, 1% antibiotic/antimycotic mix, and 1 mM HEPES (Invitrogen). For stimulation, cells were incubated with LPS-B5 (100 ng/ml, Invivogen), fMLP (1 µM, Sigma-Aldrich), PMA (100 nM, Sigma-Aldrich), and the following concentrations of recombinant mouse cytokines in combination: TNFα (100 ng/ml), IFNγ (500 U/ml), and GM-CSF (10 ng/ml), all from BioLegend. For indicated experiments, polystyrene beads (Thermo Fisher) were added to neutrophils at a 1:1 (vol:vol) ratio (MOI = 0.5). Cells were incubated with stimuli for 4 hr at 37°C and 5% $CO_2$. After incubation, cells were recovered by centrifugation, washed with PBS, and processed for flow cytometry as described above.

## Chemotaxis assay

Enriched neutrophils from the bone marrow of wild-type mice were stimulated with a cocktail of mouse recombinant cytokines (TNFα, IFNγ, GM-CSF), as described above, to induce receptor expression. After stimulation, cells were washed twice with PBS, resuspended in RPMI media supplemented with 0.1% BSA (wt/vol) to a final concentration of $1 \times 10^7$ cells/ml, and 100 µl of the cell suspension were placed in the upper compartment of a Transwell chamber (3.0 µm pore size; Corning Costar). 50 nM of recombinant mouse CCL28, CCL11 (R&D Systems), or CXCL1 (Peprotech) were placed into the lower compartment of a Transwell chamber. The Transwell plate was then incubated for 2 hr at 37°C. The number of cells that migrated to the lower compartment was determined by flow cytometry. The neutrophil chemotaxis index was calculated by dividing the number of cells that migrated in the presence of a chemokine by the number of cells that migrated in control chambers without chemokine stimulation.

## Neutrophil in vitro infection and bacterial killing assays

Bone marrow neutrophils were obtained from mice as described above. *S*. Typhimurium and *A. baumannii* were grown as described in the respective mouse experiment sections. For in vitro STm and Ab infections, bacteria were then opsonized with 20% normal mouse serum for 30 min at 37°C. After neutrophils were enriched, $1 \times 10^6$ neutrophils were seeded in a round-bottom 96-well plate with RPMI media supplemented with FBS (10%), and bacteria (STm or Ab) were added at a multiplicity of infection (MOI) = 10. The plate was centrifuged to ensure interaction between cells and bacteria, and incubated at 37°C and 5% $CO_2$. After 30 min of contact with the bacteria, the media was pipetted up and down to resuspend the cells. For analysis of CCR3 and CCR10 expression, cells were recovered at various time points (5 min, 30 min, 1 hr, 2 hr, 4 hr) by centrifugation, washed with PBS, and processed for flow cytometry as described above. For inhibition of phagocytosis, bone marrow neutrophils were pre-incubated with cytochalasin D (10 µM) in dimethyl sulfoxide (DMSO, 0.1%), or DMSO (vehicle), for 30 min prior to infection with opsonized *S*. Typhimurium for 1 hr at an MOI = 10. For killing assays, recombinant mouse CCL28 (50 nM) (*Wang et al., 2000*) and CCL11 (25 nM) (*Shamri et al., 2012*) (R&D Systems) were added to neutrophils prior to infection. When indicated, the CCR3 receptor antagonist SB328437 (Tocris Bioscience) was added at a final concentration of 10 µM (*White et al., 2000*). For assessment of bacterial killing, neutrophils infected with STm were incubated for 2.5 hr and neutrophils infected with *A. baumannii* were incubated for 4.5 hr at 37°C and 5% $CO_2$. After incubation, wells were diluted in an equal volume of PBS supplemented with 2% Triton X-100 (1% final concentration) and incubated 5 min to lyse the neutrophils, then serial dilution was performed and plated on LB agar to enumerate bacteria. To calculate the percentage of bacterial survival, the number of bacteria recovered in the presence of neutrophils was divided by the number of bacteria recovered from wells that contained no neutrophils, then multiplied by 100.

## ROS production

Neutrophils were obtained from the bone marrow of C57BL/6 wild-type mice using the EasySep Mouse Neutrophil Enrichment Kit (STEMCELL Technologies), following the manufacturer's instructions. After enrichment, $2.5 \times 10^6$ cells/ml were resuspended in phenol red-free RPMI media (Gibco) supplemented with 10% FBS (Gibco), and 1 mM HEPES (Invitrogen). The cells were incubated in presence of 2′,7′-dichlorodihydrofluorescein diacetate ($H_2DCFDA$, 25 µM) (Invitrogen), protected from light, for 30 min at 37°C and 5% of $CO_2$, as previously described (*Cao et al., 2021*). After incubation with $H_2DCFDA$, neutrophils were infected with STm as described above, then incubated for 4 hr with mouse recombinant CCL28 (50 nM), anti-mouse CCR3 antibody (5 µg/$1 \times 10^6$ cells, clone 83103), anti-mouse CCR10 antibody (5 µg/$1 \times 10^6$ cells, clone 248918), or anti-rat IgG2A (5 µg/$1 \times 10^6$ cells, clone 54447), all from R&D Systems. Neutrophils were analyzed by flow cytometry for DCF fluorescence (Ex: 492–495 nm, Em: 517–527 nm) to determine intracellular ROS production using a BD FACSCanto II flow cytometer, and data were analyzed using the FlowJo v10 software.

## NETs production

Whole-blood samples were collected from healthy donors recruited at a tertiary care center in Mexico City (Instituto Nacional de Ciencias Médicas y Nutrición Salvador Zubirán). Healthy donors signed an informed consent form before inclusion in the study, and the protocol was approved by the Instituto Nacional de Ciencias Médicas y Nutrición Salvador Zubirán ethics and research committees (Ref. 3341) in compliance with the Helsinki declaration. Neutrophils were obtained from peripheral blood of healthy voluntary donors using the EasySep Direct Human Neutrophil Isolation Kit (STEMCELL Technologies), following the manufacturer's instructions. In parallel, platelets from human peripheral blood were isolated as described (*Du et al., 2018*). Briefly, whole blood was centrifuged at $200 \times g$ for 10 min at 4°C, and plasma was recovered and then centrifuged again at $1550 \times g$ for 10 min at 4°C. The cell pellet was resuspended in RPMI media supplemented with 10% FBS ($4 \times 10^7$ cells/ml) and then incubated with LPS (5 mg/ml) for 30 min at 37°C to induce platelet activation (*Carestia et al., 2016*). For fluorescence microscopy analysis, neutrophils were incubated with autologous activated platelets (1:10 ratio) (*Liu et al., 2016*) for 3.5 hr in a 24-well plate with a poly-L-lysine-treated coverslip and stimulated with human recombinant CCL28 (50 nM) (BioLegend), the CCR3 antagonist SB328437 (10 mM, Tocris Bioscience), and/or the CCR10 antagonist BI-6901 (20 mM, Boehringer-Ingelheim). Cells were then incubated with the DNA-binding dye Helix-NP Green (10 nM, BioLegend) for 30 min, and then fixed with paraformaldehyde (2%). Coverslips were mounted in slides using a mounting medium with DAPI (Fisher Scientific), and images were taken with a fluorescence microscope (Zeiss). At least 3 fields per sample were analyzed to determine the percentage of cells forming NETs. For flow cytometry analysis, neutrophils were stimulated for 2.5 hr as described above, and then incubated with the dye Helix-NP and human anti-MPO-Biotin antibody (clone MPO421-8B2, Novus Biologicals), and APC/Cy7 streptavidin (BioLegend). Samples were analyzed using an LSR Fortessa flow cytometer (BD Biosciences) to determine the presence of DNA–MPO complexes (*Masuda et al., 2017*), and data were analyzed using FlowJo v10 software.

## Growth of bacteria in media supplemented with recombinant chemokines

*S.* Typhimurium wild-type, *S.* Typhimurium *phoQ* mutant, and *Escherichia coli* K12 were grown in LB broth overnight at 37°C. *A. baumannii* was cultured in Cation-Adjusted Mueller-Hinton Broth (CA-MHB) under the same conditions. The following day, cultures were diluted 1:100 in LB and grown at 37°C for 3 hr, subsequently diluted to $\sim 0.5 \times 10^6$ or $\sim 0.5 \times 10^9$ CFU/ml in 1 mM potassium phosphate buffer (pH 7.2), then incubated at 37°C in the presence or absence of recombinant murine CCL28 (BioLegend) at the indicated concentrations. After 2 hr, samples were plated onto LB agar to enumerate viable bacteria. In other assays, *S.* Typhimurium was grown as described above and $\sim 1 \times 10^7$ CFU/ml were incubated at 37°C for 2.5 hr in the presence or absence of recombinant murine CCL28 (50 nM) (*Wang et al., 2000*) or CCL11 (25 nM) (*Shamri et al., 2012*) in RPMI medium supplemented with 10% FBS. After incubation, samples were plated onto LB + Nal agar to enumerate viable bacteria.

## RNA extraction and qPCR

Total RNA was extracted from mouse cecal or lung tissue using Tri-Reagent (Molecular Research Center). Reverse transcription of 1 µg of total RNA was performed using the SuperScript VILO cDNA Synthesis kit (Thermo Fisher Scientific). Quantitative real-time PCR for the expression of *Actb* (β-actin), *Cxcl1*, *Tnfa*, *Ifng*, *Csf3*, *Il1b*, and *Il17a* was performed using the PowerUp SYBR Green Master Mix (Applied Biosystems) on a QuantStudio 5 Real-Time PCR System (Thermo Fisher Scientific). Gene expression was normalized to *Actb* (β-actin). Fold changes in gene expression were relative to average expression in uninfected controls and calculated using the ΔΔCt method.

## Histopathology

Cecal and lung tissue samples collected at necropsy were fixed in 10% buffered formalin for 24–48 hr, then transferred to 70% ethanol for storage. Tissues were embedded in paraffin according to standard procedures and sectioned at 5 µm. Pathology scores of cecal and lung samples were determined by blinded examinations of hematoxylin and eosin-stained sections. Each cecal section was evaluated using a semiquantitative score as described previously (*Moschen et al., 2016*). Lung inflammation was assessed by a multiparametric scoring based on previous work (*Lammers et al., 2012*).

## Immunofluorescence

Deparaffinized lung sections were stained with a purified rat anti-mouse Ly6G antibody (clone 1A8, BioLegend) according to standard immunohistochemical procedures. Ly6G$^+$ cells were visualized by a goat anti-rat secondary antibody (Invitrogen). Cell nuclei were stained with DAPI in SlowFade Gold Antifade Mountant (Invitrogen). Slides were scanned on a Zeiss Axio Scan.Z1 slide scanner and whole lung scans were evaluated with QuPath analysis software (*Bankhead et al., 2017*). Ly6G$^+$ cells per mouse were quantified by averaging the neutrophil numbers of three consecutive high-power fields in regions with moderate to severe inflammation.

## Statistical analysis

Statistical analysis was performed with GraphPad Prism 10. CFU data from in vivo infection experiments, percentage of CCR3$^+$ or CCR10$^+$ neutrophils in vivo and in vitro, and data from neutrophil functional assays were transformed to Log10 and passed a normal distribution test before running statistical analyses. Data on cytokine secretion, qPCR data, and relative cell abundances within tissues were compared by Mann–Whitney *U* test. Survival curves were compared by the Log-rank (Mantel–Cox) test. Data that were normally distributed were analyzed by one-way analysis of variance (ANOVA) for independent samples or paired samples, depending on the experimental setup. Dunnett's multiple comparisons test was applied when we compared the different conditions to a single control group, while Tukey's multiple comparison test was performed when we compared each condition with each other. Greenhouse–Geisser correction was applied when there were differences in the variance among the groups. Data from chemokine migration were analyzed by a non-parametric ANOVA (Kruskal–Wallis's test), assuming non-equal standard deviation given the differences in the variance among the groups and followed by Dunn's multiple comparisons test. Paired *t* test was used when only two paired experimental groups were compared. A p value equal to or below 0.05 was considered statistically significant. * indicates an adjusted p-value ≤0.05, p-value ≤0.01, p-value ≤0.001, p-value ≤0.0001.

## Acknowledgements

This work was supported by the NIH (Public Health Service Grants AI121928) to MR, by a pilot project award from the NIAID Mucosal Immunology Studies Team (MIST) to APL, and by a grant from the InnovaUNAM of the National Autonomous University of Mexico (UNAM) and Alianza UCMX of the University of California, to APL and MR. RRG was partly supported by a fellowship from the Max Kade Foundation and by a fellowship from the Crohn's and Colitis Foundation. MHL was partly supported by NIH training grant T32 DK007202 and by NIH F32 AI169989. ND was supported by NIH training grant NIH 5T32HD087978-05 and NIH NIAID grant 1-U01-AI124316. GTW was supported by NIH training grant T32AI007036. KM was partly supported by São Paulo Research Foundation (FAPESP) Grants 2019/14833-0 and 2018/22042-0. RCD, SR-R, and VAS-H were supported by a Grad School Fellowship from CONACyT. MR and VN were supported by Public Health Service Grant AI145325.

Work in JLM-M lab was supported by CONACyT-FOSISS grant A3-S-36875 and UNAM-DGAPA-PAPIIT Program grant IN213020. Work in MR lab was also supported by NIH Grants AI126277, AI154644, AI096528, by AMED grant JP233fa627003, and by the Chiba University-UCSD Center for Mucosal Immunology, Allergy, and Vaccines, and by an Investigator in the Pathogenesis of Infectious Disease Award from the Burroughs Wellcome Fund. We would like to thank Dr. Albert Zlotnik for his thoughtful suggestions on the project over the years. We also acknowledge Boehringer-Ingelheim Pharma GmbH & Co. KG for the kind gift of the CCR10 antagonist BI-6901, the Histology Core at the La Jolla Institute for Immunology, and the support from the San Diego Digestive Diseases Research Center (P30 DK120515).

## Additional information

### Funding

| Funder | Grant reference number | Author |
| --- | --- | --- |
| National Institute of Allergy and Infectious Diseases | AI121928 | Manuela Raffatellu |
| National Institute of Allergy and Infectious Diseases | Mucosal Immunology Studies Team | Araceli Perez-Lopez |
| Crohn's and Colitis Foundation | 649744 | Romana R Gerner |
| National Institute of Diabetes and Digestive and Kidney Diseases | DK007202 | Michael H Lee |
| National Institute of Allergy and Infectious Diseases | AI169989 | Michael H Lee |
| Eunice Kennedy Shriver National Institute of Child Health and Human Development | HD087978 | Nicholas Dillon |
| National Institute of Allergy and Infectious Diseases | AI124316 | Nicholas Dillon Victor Nizet |
| National Institute of Allergy and Infectious Diseases | AI007036 | Gregory T Walker |
| National Institute of Allergy and Infectious Diseases | AI145325 | Victor Nizet |
| Japan Agency for Medical Research and Development | JP233fa627003 | Manuela Raffatellu |
| Burroughs Wellcome Fund | | Manuela Raffatellu |
| National Institute of Diabetes and Digestive and Kidney Diseases | DK120515 | Manuela Raffatellu |
| National Autonomous University of Mexico | InnovaUNAM grant | Araceli Perez-Lopez Manuela Raffatellu |
| University of California | Alianza UCMX | Araceli Perez-Lopez Manuela Raffatellu |
| Max Kade Foundation | Fellowship | Romana R Gerner |
| São Paulo Research Foundation | 2019/14833- 0 | Karine Melchior |
| São Paulo Research Foundation | 2018/22042- 0 | Karine Melchior |

| Funder | Grant reference number | Author |
|---|---|---|
| Consejo Nacional de Ciencia y Tecnología | Grad School Fellowship | Rodrigo Cervantes-Diaz Sandra Romero-Ramirez Victor A Sosa-Hernandez |
| Public Health Service Grant | AI145325 | Manuela Raffatellu Victor Nizet |
| Consejo Nacional de Ciencia y Tecnología | CONACyT- FOSISS A3-S-36875 | Jose L Maravillas-Montero |
| Universidad Nacional Autónoma de México | UNAM- DGAPA- PAPIIT Program grant IN213020 | Jose L Maravillas-Montero |

The funders had no role in study design, data collection, and interpretation, or the decision to submit the work for publication.

## Author contributions

Gregory T Walker, Conceptualization, Data curation, Formal analysis, Supervision, Validation, Investigation, Visualization, Methodology, Writing – original draft, Writing – review and editing; Araceli Perez-Lopez, Conceptualization, Data curation, Formal analysis, Supervision, Funding acquisition, Validation, Investigation, Visualization, Methodology, Writing – original draft, Writing – review and editing, Conceived the overall study; Steven Silva, Formal analysis, Investigation, Visualization, Methodology; Michael H Lee, Formal analysis, Supervision, Investigation, Visualization, Methodology, Writing – review and editing; Elisabet Bjånes, Grant J Norton, Felix A Argueta, Frenchesca Dela Pena, Lauren Park, Victor A Sosa-Hernandez, Rodrigo Cervantes-Diaz, Sandra Romero-Ramirez, Investigation, Methodology; Nicholas Dillon, Stephanie L Brandt, Investigation, Methodology, Writing – review and editing; Romana R Gerner, Formal analysis, Investigation, Visualization, Methodology, Writing – review and editing, Analyzed the histopathology; Karine Melchior, Investigation, Visualization, Methodology; Monica Cartelle Gestal, Supervision, Methodology; Jose L Maravillas-Montero, Supervision, Funding acquisition, Investigation, Methodology, Writing – review and editing; Sean-Paul Nuccio, Resources, Data curation, Supervision, Writing – original draft, Writing – review and editing; Victor Nizet, Conceptualization, Supervision, Funding acquisition, Writing – original draft, Writing – review and editing; Manuela Raffatellu, Conceptualization, Data curation, Supervision, Funding acquisition, Writing – original draft, Project administration, Writing – review and editing, Conceived the overall study

## Author ORCIDs

Gregory T Walker ⓘ http://orcid.org/0000-0002-3888-8143
Araceli Perez-Lopez ⓘ https://orcid.org/0000-0002-5399-1958
Sean-Paul Nuccio ⓘ https://orcid.org/0000-0002-9683-9278
Victor Nizet ⓘ https://orcid.org/0000-0003-3847-0422
Manuela Raffatellu ⓘ https://orcid.org/0000-0001-6487-4215

## Ethics

Whole-blood samples were collected from healthy donors recruited at a tertiary care center in Mexico City (Instituto Nacional de Ciencias Médicas y Nutrición Salvador Zubirán). Healthy donors signed an informed consent form before inclusion in the study, and the protocol was approved by the Instituto Nacional de Ciencias Médicas y Nutrición Salvador Zubirán ethics and research committees (Ref. 3341) in compliance with the Helsinki declaration.
All mouse experiments were reviewed and approved by the Institutional Animal Care and Use Committees at UC Irvine (protocol #2009-2885) and UC San Diego (protocols #S17107 and #S00227M).

## Decision letter and Author response

Decision letter https://doi.org/10.7554/eLife.78206.sa1
Author response https://doi.org/10.7554/eLife.78206.sa2

## Additional files

### Supplementary files
• Transparent reporting form

### Data availability
All data generated or analyzed during this study are included in the manuscript and supporting files. Raw data are available at Dryad.

The following dataset was generated:

| Author(s) | Year | Dataset title | Dataset URL | Database and Identifier |
|---|---|---|---|---|
| Raffatellu M | 2024 | Data from: CCL28 modulates neutrophil responses during infection with mucosal pathogens | https://dx.doi.org/10.5061/dryad.59zw3r2j6 | Dryad Digital Repository, 10.5061/dryad.59zw3r2j6 |

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
