## [Editor Report]

This important study provides compelling evidence that CCL28 plays a crucial role in regulating neutrophil function and host defense during mucosal infections, with CCL28 deficiency leading to greater susceptibility to *Salmonella* gut infections and enhanced resistance to Acinetobacter lung infections. The data convincingly shows that CCL28, through CCR3, regulates neutrophil functions such as reactive oxygen species production and extracellular trap formation, influencing pathogen clearance and highlighting its context-dependent impact on immunity.

---

## [Decision Letter]

**Decision letter after peer review:**

Thank you for submitting your article "CCL28 modulates neutrophil responses during infection with mucosal pathogens" for consideration by *eLife*. Your article has been reviewed by 3 peer reviewers, and the evaluation has been overseen by a Reviewing Editor and Carla Rothlin as the Senior Editor. The following individual involved in the review of your submission has agreed to reveal their identity: Denise M Monack (Reviewer #2).

Essential revisions (for the authors):

1) A more detailed examination of CCR3 and CCR10 surface translocation in neutrophils, including the kinetics and rapidity (Reviewer #1); What triggers this, including analysis of additional stimuli such as Acinetobacter or CCL28 (Reviewers #2 and #3). In addition, are CCR3 mRNA or protein levels induced in neutrophils by stimuli (Reviewer #1).

2) Deeper analysis of immune cells in the Ccl28-/- mice, including eosinophils in the gut and lungs, and neutrophil phenotypes (e.g. AMP levels) (Reviewer #2).

3) Deeper analysis of NETosis phenotypes and mechanisms for neutrophils in response to CCL28 or in Ccl28-/- mice (Reviewer #2).

4) Better explanation/justification for specific time points used for STm infection experiments, and clarification of in vivo phenotypes for neutrophils that were reported (Reviewer #3).

*Reviewer #1 (Recommendations for the authors):*

1. The authors show by ICS that CCR3 is already preformed in neutrophils, and their data indicate that CCR3 translocates to the plasma membrane. How rapid is this translocation of CCR3 to the cell surface? The authors examined 4 hours post-stimulation – did they also examine earlier timepoints? In addition, are Ccr3 mRNA and total protein levels also inducible in neutrophils in response to various stimuli?

*Reviewer #2 (Recommendations for the authors):*

Specific Comments:

1) The authors acknowledge that CCL28 are involved in the recruitment of T-, B-cells and eosinophils. While the authors provide data on lymphocyte populations in their Ccl28-/- mice compared to wild-type mice, there is no data regarding any potential effects on eosinophil populations in the gut or the lungs. As eosinophils are known to induce lung pathology, it would be good to know if these cells are influencing the phenotypes observed in their infection models.

2) When does CCL28 signalling become important in the neutrophil response? If surface receptor expression needs to be induced by other cytokines/chemokines, is it possible that there is a secondary role of CCL28 in vivo.

3) There is a general lack of description of the neutrophil phenotype in the Ccl28-/- mice. Do Ccl28-/- neutrophils contain the same amount AMPs and cytokine as WT neutrophils and are there differences in their degranulation and NETosis ability?

4) In line with the previous question, do CCL28 trigger the release of mitochondrial or nuclear NETs? It would also be advisable to use additional methods to quantify or visualise NETs, either by microscopy or by the SYTOX-green NETs assay.

5) Would the authors expect a similar phenotype in their models if they depleted neutrophils using antibodies/neutrophil deficient mice?

6) The authors write and reference that CCL28 is known to be released by epithelial cells in the gut and in the lung. It would be nice to see if there are additional cell types involved in the CCL28 response.

7) It would be nice to identify triggers for the translocation of CCR3 and CCR10 from the intracellular compartment to the extracellular compartment.

*Reviewer #3 (Recommendations for the authors):*

1) Why did the authors look for CCL28 levels by ELISA in the feces and not in the cecal/colonic mucosa?

2) Figure 1B: Legend for Y-axis should be included in all figure panels.

3) Lines 106-108: authors should clarify which time-point was used for the intraperitoneal S. Tm experiment.

4) Why did the authors decide to use 48h time-point to characterize infiltration of inflammatory cells during S. Tm infection by flow cytometry (Figures 1C-D)? A different time point (72h post-infection) was used in previous experiments (Figure 1A-B).

5) S. Tm benefits from intestinal inflammation to expand in the intestinal lumen. However, CCL28-/- mice have a higher S. Tm burden in cecal contents (Figure 1B) even though they show decreased intestinal inflammation (Figure 1F). How do authors explain the data that S. Tm can colonize the gut better when there is less intestinal inflammation?

6) Line 164: authors should clarify which time-point was used for Acinetobacter CFU counts in bronchoalveolar lavage (BAL) fluid or blood (Figure 2B, C).

7) In line 213, the authors state that "Ccl28-/- mice expressed similar levels of these receptors as wild-type mice." – Shouldn't Ccl28-/- have lower levels of neutrophils that express the CCL28 receptors (CCR3, CCR10) when compared to WT mice, as these neutrophils wouldn't be attracted to the gut mucosa due to the absence of CCL28? The similar levels of CCR3 and CCR10 expressing neutrophils between WT and Ccl28-/- should be better explained by the authors.

8) Does Acinetobacter also induce CCR3 and CCR10 expression neutrophils (as shown for S. Tm)? On a similar note, does exposure of neutrophils to CCL28 alter CCR3 and CCR10 expression or their localization to cell surface?

---

## [Author Response]

Essential revisions (for the authors):1) A more detailed examination of CCR3 and CCR10 surface translocation in neutrophils, including the kinetics and rapidity (Reviewer #1); What triggers this, including analysis of additional stimuli such as Acinetobacter or CCL28 (Reviewers #2 and #3). In addition, are CCR3 mRNA or protein levels induced in neutrophils by stimuli (Reviewer #1).

We agree that providing a more detailed examination of CCR3 and CCR10 surface expression in neutrophils is important. To this end, we have performed new experiments in which neutrophils enriched from bone marrow were infected with *Salmonella* Typhimurium (STm) or *Acinetobacter baumannii* (Ab) for 5 min, 30 min, 1 hr, 2 hr, or 4 hr (Figure 4A-D and Figure 4—figure supplement 1A-D).

STm induced a rapid surface expression of CCR3 beginning 5 min post-infection (p.i.), which increased over time and was maximal at 2-4 hr p.i. (Figure 4A). CCR10 was expressed on the surface of only a small percentage of neutrophils and increased to up to ~1% 4 hr after STm infection (Figure 4—figure supplement 1A), when also intracellular CCR10 expression was the highest (~20%; Figure 4—figure supplement 1B). When neutrophils were infected with Ab, CCR3 induction was slower, starting at 1 hr p.i., and peaking at 2-4 hr p.i. (Figure 4C). In contrast, Ab infection did not significantly increase surface and intracellular CCR10 expression (Figure 4—figure supplement 1C-D).

Because we saw the most robust induction of surface CCR3 (and surface CCR10 to a lesser extent) during STm infection, we performed RNA extraction from uninfected neutrophils and neutrophils 1 hr after STm infection. We found that *Ccr3* and *Ccr10* mRNA were induced in neutrophils infected with STm relative to uninfected neutrophils, but the induction level was low (~1.8-2-fold). We provide these data in response to Reviewer 1’s point #1.

2) Deeper analysis of immune cells in the Ccl28-/- mice, including eosinophils in the gut and lungs, and neutrophil phenotypes (e.g. AMP levels) (Reviewer #2).

We agree that providing a deeper analysis of the immune cells in *Ccl28*^-/-^ mice is important. To address this point, we needed to conduct extensive mouse breeding to repeat, in essence, our prior study and isolate live cells for more comprehensive immune phenotyping. We compared naive mice to mice infected with STm at both day 2 and day 3 p.i., which also enabled us to address point #4 below. We also compared naive mice to mice infected with Ab for 1 day. Our flow cytometry strategy for comprehensive immune phenotyping is shown in the Figure 1—figure supplement 3, and the results are shown in Figure 1D,E, Figure 2I-L, Figure 1—figure supplements 4 and 5, and Figure 2—figure supplements 1 and 2.

During STm infection, we detected fewer neutrophils in the gut of *Ccl28*^-/-^ mice at both day 2 and day 3 p.i. (new Figure 1D,E). At the same time points, we did not find differences in the frequency of other cell types, including eosinophils, macrophage-like F4/80^+^ CD11c^-^ cells, conventional dendritic cell-like CD11c^+^ F4/80^-^ cells, B cells, CD4^+^ T cells, CD8^+^ T cells, or CD4^+^ CD8^+^ cells in the gut, blood or bone marrow (Figure 1—figure supplement 4 and Figure 1—figure supplement 5). In essence, the new immune phenotyping further supports our initial findings, showing that during STm infection, *Ccl28*^-/-^ mice are characterized primarily by a lower percentage of neutrophils in the gut.

In mice infected with Ab, most cells (~90%) in the BAL fluid were neutrophils. Even though the percentage of BAL fluid neutrophils was comparable between WT and *Ccl28*^-/-^ mice, the number of neutrophils was decreased in the BAL fluid (Figure 2I-L). The percentage and abundance of other BAL fluid cells, including eosinophils, were comparable between WT and *Ccl28*^-/-^ mice (Figure 2L, Figure 2figure supplement 1 and Figure 2—figure supplement 2). No differences were observed in any other tissue analyzed (lung, blood, bone marrow; Figure 2—figure supplements 1 and 2). Also in this case, deeper analysis of the cellular infiltrate during Ab infection is consistent with our initial findings and demonstrates that *Ccl28*^-/-^ mice have fewer neutrophils in the BAL fluid.

To test for possible differences in the neutrophil phenotypes between wild-type and *Ccl28*^-/-^ mice, we measured levels of the neutrophil markers elastase, myeloperoxidase, and calprotectin by ELISA in the fecal or cecal content of mice at day 2 and 3 after STm infection, and in the BAL fluid of mice at day 1 after Ab infection. We found that there was a trend towards lower levels of these neutrophil markers in the cecal content at day 3 after STm infection, and in the BAL fluid at day 1 after Ab infection, in *Ccl28*^-/-^ mice (Figure 2—figure supplement 3). These differences likely reflect the lower neutrophil abundance in *Ccl28*^-/-^ mice, rather than changes in expression of these neutrophil proteins. In future studies outside the scope of this manuscript, we plan to further characterize the neutrophils from infected WT and *Ccl28*^-/-^ mice.

3) Deeper analysis of NETosis phenotypes and mechanisms for neutrophils in response to CCL28 or in Ccl28-/- mice (Reviewer #2).

To provide a deeper analysis of NETosis phenotypes, we have expanded on our study in plateletactivated human neutrophils stimulated with CCL28, and we have visualized NET formation by fluorescence microscopy. The new results show that CCL28 enhances NET formation, and that inhibition of CCR3 results in a reduction of NETosis (Figure 5H-I).

4) Better explanation/justification for specific time points used for STm infection experiments, and clarification of in vivo phenotypes for neutrophils that were reported (Reviewer #3).

In the first version of the manuscript, we used different time points based on the severity of infection, which varied between the two animal facilities at UC Irvine (experiments until 2017) and UCSD (experiments after 2017, due to the Raffatellu lab’s relocation to UCSD). As we have now performed several additional experiments necessary to address the deeper analysis of the immune cells requested by the reviewers, we took the opportunity to collect additional data on both day 2 and day 3 after STm infection. In general, day 3 post-infection shows higher levels of inflammation and more significant differences in the CFUs and the neutrophil infiltrate. For simplicity, we have now included the data from day 3 p.i. in the main figures, whereas most data from day 2 p.i. is shown in the Supplemental figures.

Reviewer #1 (Recommendations for the authors):1. The authors show by ICS that CCR3 is already preformed in neutrophils, and their data indicate that CCR3 translocates to the plasma membrane. How rapid is this translocation of CCR3 to the cell surface? The authors examined 4 hours post-stimulation – did they also examine earlier timepoints? In addition, are Ccr3 mRNA and total protein levels also inducible in neutrophils in response to various stimuli?

The reviewer raises a very important point about the time course of CCR3 surface expression. To address this, we have performed new experiments in which neutrophils enriched from bone marrow were infected with *Salmonella* Typhimurium (STm) or *Acinetobacter baumannii* (Ab) for 5 min, 30 min, 1 hr, 2 hr, or 4 hr (Figure 4A-D and Figure 4—figure supplement 1A-D).

STm induced a rapid surface expression of CCR3 beginning 5 min post-infection (p.i.), which increased over time and was maximal at 2-4 hr p.i. (Figure 4A). CCR10 was expressed on the surface of only a small percentage of neutrophils and increased to up to ~1% 4 hr after STm infection (Figure 4—figure supplement 1A), when also intracellular CCR10 expression was the highest (~20%; Figure 4—figure supplement 1B). When neutrophils were infected with Ab, CCR3 induction was slower, starting at 1 hr p.i., and peaking at 2-4 hr p.i. (Figure 4C). In contrast, Ab infection did not significantly increase surface and intracellular CCR10 expression (Figure 4—figure supplement 1C,D).

Because we saw the most robust induction of surface CCR3 (and surface CCR10 to a lesser extent) during STm infection, we performed RNA extraction from uninfected neutrophils and neutrophils 1 hr after STm infection. We found that *Ccr3* and *Ccr10* mRNA were induced in neutrophils infected with STm relative to uninfected neutrophils, but the induction level was low (~1.8-2-fold) (Author response image 1). As we find that virtually all neutrophils are positive for intracellular CCR3, and because of how rapidly it localized to the cell surface (within 5 min), we think that pre-formed CCR3 is the primary source of surface CCR3. However, as it is possible that newly produced CCR3 also contributes to surface CCR3, we have toned down this claim in the manuscript.

**Author response image 1. sa2fig1:** *Ccr3* and *Ccr10* expression is induced in neutrophils by STm in vivo. Enriched bone marrow neutrophils (1x10^6^ cells/well) were infected in vitro with NMS-opsonized STm (MOI=10) or mock-infected with vehicle (DPBS+10%NMS) for 1 hr. Cells were then resuspended in Tri-Reagent, total RNA was extracted, and reverse-transcribed into cDNA. qRT-PCR was performed for the expression of *Ccr3* and *Ccr10*, normalized to the expression of *Actb*. Change in expression in STm-infected cells was normalized to relative expression in the uninfected control cells. Symbols represent data from bone marrow cells extracted from individual mice, lines represent the mean.

Reviewer #2 (Recommendations for the authors):Specific Comments:1) The authors acknowledge that CCL28 are involved in the recruitment of T-, B-cells and eosinophils. While the authors provide data on lymphocyte populations in their Ccl28-/- mice compared to wild-type mice, there is no data regarding any potential effects on eosinophil populations in the gut or the lungs. As eosinophils are known to induce lung pathology, it would be good to know if these cells are influencing the phenotypes observed in their infection models.

We agree that providing a deeper analysis of the immune cells in *Ccl28*^-/-^ mice is important. To address this point, we needed to conduct extensive mouse breeding to repeat, in essence, our prior study and isolate live cells for more comprehensive immune phenotyping. We compared naive mice to mice infected with STm at both day 2 and day 3 p.i., which also enabled us to address point #4 below. We also compared naive mice to mice infected with Ab for 1 day. Our flow cytometry strategy for comprehensive immune phenotyping is shown in the Figure 1—figure supplement 3, and the results are shown in Figure 1D,E, Figure 2 I-L, Figure 1—figure supplements 4 and 5, and Figure 2—figure supplements 1 and 2.

During STm infection, we detected fewer neutrophils in the gut of *Ccl28*^-/-^ mice at both day 2 and day 3 p.i. (new Figure 1D,E). At the same time points, we did not find differences in the frequency of other cell types, including eosinophils, macrophage-like F4/80^+^ CD11c^-^ cells, conventional dendritic cell-like CD11c^+^ F4/80^-^ cells, B cells, CD4^+^ T cells, CD8^+^ T cells, or CD4^+^ CD8^+^ cells in the gut, blood or bone marrow (Figure 1—figure supplements 4 and 5). In essence, the new immune phenotyping further supports our initial findings, showing that during STm infection, *Ccl28*^-/-^ mice are characterized primarily by a lower percentage of neutrophils in the gut.

In mice infected with Ab, most cells (~90%) in the BAL fluid were neutrophils. Even though the percentage of BAL fluid neutrophils was comparable between WT and *Ccl28*^-/-^ mice, the number of neutrophils was decreased in the BAL fluid (Figure 2I-L). The percentage and abundance of other BAL fluid cells, including eosinophils, were comparable between WT and *Ccl28*^-/-^ mice (Figure 2L, and Figure 2—figure supplements 1 and 2). No differences were observed in any other tissue analyzed (lung, blood, bone marrow; Figure 2—figure supplements 1 and 2). Also in this case, deeper analysis of the cellular infiltrate during Ab infection is consistent with our initial findings and demonstrate that *Ccl28*^-/-^ mice have fewer neutrophils in the BAL fluid.

In summary, the extensive immunophenotyping in two infection models further strengthened our conclusions that *Ccl28*^-/-^ mice primarily exhibit a reduction in neutrophil recruitment to the gut during STm infection, and to the lung during Ab infection.

2) When does CCL28 signalling become important in the neutrophil response? If surface receptor expression needs to be induced by other cytokines/chemokines, is it possible that there is a secondary role of CCL28 in vivo.

Indeed, we find that other proinflammatory stimuli (bacterial infection, cytokines, phagocytosis) are essential for inducing surface receptor expression (Figure 3). We agree that CCL28 can also play secondary roles in addition to modulating neutrophil responses. CCL28 is thought to have both inflammatory and homeostatic functions, many of which are still unknown as *Ccl28*^-/-^ mice have been developed only recently. As pointed out by Reviewer 1, CCL28 may also promote B and T cell responses in chronic infection models. We now discuss that future studies, outside the scope of the current manuscript, are necessary to elucidate additional roles for CCL28.

3) There is a general lack of description of the neutrophil phenotype in the Ccl28-/- mice. Do Ccl28-/- neutrophils contain the same amount AMPs and cytokine as WT neutrophils and are there differences in their degranulation and NETosis ability?

To test for possible differences in the neutrophil phenotypes between wild-type and *Ccl28*^-/-^ mice, we measured levels of the neutrophil markers elastase, myeloperoxidase, and calprotectin by ELISA in the fecal or cecal content of mice at day 2 and 3 after STm infection, and in the BAL fluid of mice at day 1 after Ab infection. We found that there was a trend towards lower levels of these neutrophil markers in the cecal content at day 3 after STm infection, and in the BAL fluid at day 1 after Ab infection, in *Ccl28*^-/-^ mice (Figure 1—figure supplement 6). These differences likely reflect the lower neutrophil abundance in *Ccl28*^-/-^ mice, rather than changes in expression of these neutrophil proteins. In future studies outside the scope of this manuscript, we plan to further characterize the neutrophils from infected WT and *Ccl28*^-/-^ mice.

4) In line with the previous question, do CCL28 trigger the release of mitochondrial or nuclear NETs? It would also be advisable to use additional methods to quantify or visualise NETs, either by microscopy or by the SYTOX-green NETs assay.

We have now performed fluorescence microscopy using the Helix dye (same as SYTOX-green, but sold by BioLegend) in platelet-activated human neutrophils. We analyzed at least 3 fields per sample by fluorescence microscopy to identify and determine the percentage of cells forming NETs. The new images and the quantifications are shown in Figure 5H-I. These results are consistent with the prior flow cytometry results (now shown in Figure 5—figure supplement 1) and confirm that CCL28 induces NET formation, and that inhibition of CCR3 results in a reduction of NETosis.

The use of Helix dye does not allow discrimination between mitochondrial and nuclear NETs. However, prior work has shown that activated platelets induce nuclear NETs in human neutrophils (PMID 22684106), suggesting that nuclear NETs are present in our experimental settings. Still, we can not rule out the possibility that CCL28 induces mitochondrial NETs. Further characterization of NETs will be the investigation of future studies outside the scope of this manuscript.

5) Would the authors expect a similar phenotype in their models if they depleted neutrophils using antibodies/neutrophil deficient mice?

Based on the literature and our experience, we expect neutrophil-deficient mice, or mice treated with antibodies to deplete neutrophils, to develop more severe disease than *Ccl28*^-/-^ mice during STm infection and Ab infection. It is important to note that *Ccl28*^-/-^ mice exhibit a significant reduction of neutrophils in the infected mucosal sites, but not a complete depletion. Moreover, *Ccl28*^-/-^ mice have normal neutrophil numbers in other body sites, including blood and bone marrow. The suggested antibody depletion studies would deplete neutrophils also from blood and bone marrow. Therefore, we expect that mice with a complete absence/depletion of neutrophils from all body sites will exhibit a much more severe disease during infection.

6) The authors write and reference that CCL28 is known to be released by epithelial cells in the gut and in the lung. It would be nice to see if there are additional cell types involved in the CCL28 response.

It is possible that other cell types secrete CCL28 in the gut and lung in addition to epithelial cells, and that other cell types may be involved in the CCL28 response. Teasing out these additional responses would require extensive additional experiments outside the current study's scope and will be the subject of future studies.

7) It would be nice to identify triggers for the translocation of CCR3 and CCR10 from the intracellular compartment to the extracellular compartment.

To address a comment from Reviewer 1, we have performed time-course experiments showing that CCR3 surface expression can be detected within 5 minutes after STm infection while, for example, the response to Ab is slower (Figure 4A-D). We plan to investigate the molecular mechanisms in future studies, which are beyond the scope of this manuscript.

Reviewer #3 (Recommendations for the authors):1) Why did the authors look for CCL28 levels by ELISA in the feces and not in the cecal/colonic mucosa?

We thank the reviewer for this question. In prior studies, CCL28 was detected in the colonic mucosa of humans. We attempted to detect CCL28 by immunohistochemistry in mouse tissue but were unsuccessful. As antibodies specific for IHC in mice were not available, we used the LSBio polyclonal rabbit anti-Human CCL28/MEC antibody (LS-C379500) for IHC, which is described to cross-react with mice. We found the antibody to stain wild-type and *Ccl28^-/-^* gut tissues non-specifically, as we also detected a positive signal in *Ccl28*^-/-^ mice.

2) Figure 1B: Legend for Y-axis should be included in all figure panels.

We agree and we have now changed this in all relevant figures.

3) Lines 106-108: authors should clarify which time-point was used for the intraperitoneal S. Tm experiment.

We agree with the reviewer. The time point was reported in the figure legend and the Materials and methods, but not in the main text. We have now clarified in the main text that mice were infected intraperitoneally for 4 days.

4) Why did the authors decide to use 48h time-point to characterize infiltration of inflammatory cells during S. Tm infection by flow cytometry (Figures 1C-D)? A different time point (72h post-infection) was used in previous experiments (Figure 1A-B).

The reviewer raises an important point. In the first version of the manuscript, we used different time points based on the severity of infection, which varied between the two animal facilities at UC Irvine (experiments until 2017) and UCSD (experiments after 2017, due to the Raffatellu lab relocation to UCSD). As we have now performed several additional experiments necessary to address the deeper analysis of the immune cells requested by the reviewers, we took the opportunity to collect additional data on both day 2 and day 3 after STm infection. In general, day 3 post-infection shows higher levels of inflammation and more significant differences in the CFUs and the neutrophil infiltrate. For simplicity, we have now included the data from day 3 p.i. in the main figures, whereas most data from day 2 p.i. are shown in the Supplemental figures.

5) S. Tm benefits from intestinal inflammation to expand in the intestinal lumen. However, CCL28-/- mice have a higher S. Tm burden in cecal contents (Figure 1B) even though they show decreased intestinal inflammation (Figure 1F). How do authors explain the data that S. Tm can colonize the gut better when there is less intestinal inflammation?

As discussed above, we have now performed several additional experiments, and we have enumerated STm CFUs in a larger number of mice. By combining data from four independent experiments (WT, n=14; *Ccl28*^-/-^, n=13), we find that *Ccl28*^-/-^ mice have lower STm CFUs in the cecal content at day 2 post-infection (Figure 1—figure supplement 1A), but have comparable STm CFU levels to WT mice at day 3 post-infection (eight independent experiment (WT, n=24; *Ccl28*^-/-^, n=18); Figure 1B). In general, fecal colonization was similar between WT and *Ccl28*^-/-^ mice at all time points analyzed (Figure 1B and Figure 1—figure supplement 1A).

6) Line 164: authors should clarify which time-point was used for Acinetobacter CFU counts in bronchoalveolar lavage (BAL) fluid or blood (Figure 2B, C).

We have clarified in the main text that tissue samples for Ab CFU counts were collected at day 1 postinfection.

7) In line 213, the authors state that "Ccl28-/- mice expressed similar levels of these receptors as wild-type mice." – Shouldn't Ccl28-/- have lower levels of neutrophils that express the CCL28 receptors (CCR3, CCR10) when compared to WT mice, as these neutrophils wouldn't be attracted to the gut mucosa due to the absence of CCL28? The similar levels of CCR3 and CCR10 expressing neutrophils between WT and Ccl28-/- should be better explained by the authors.

We thank the reviewer for this comment. We would like to clarify that we see similar frequencies of CCR3^+^ and CCR10^+^ neutrophils in WT and *Ccl28*^-/-^ mice (Figure 3—figure supplement 1). However, *Ccl28*^-/-^ mice indeed have fewer neutrophils in the mucosa during infection (shown, for instance, in Figure 1D-E for STm infection, and Figure 2G-H for Ab infection). Thus, these results are consistent with CCL28-dependent modulation of neutrophil trafficking to the mucosa. Expression of the receptors is independent of CCL28, but dependent on other stimuli and can happen very rapidly during infection (see also the new timecourse data for neutrophils infected with STm or Ab shown Figure 4 A-D and Figure 4—figure supplement 1A-D). We hope that the revised manuscript and figures better convey the results.

8) Does Acinetobacter also induce CCR3 and CCR10 expression neutrophils (as shown for S. Tm)? On a similar note, does exposure of neutrophils to CCL28 alter CCR3 and CCR10 expression or their localization to cell surface?

In vitro, neutrophils infected with Ab displayed CCR3 induction, albeit slower than with STm, starting at 1 hr p.i. (versus 5 min for STm), and peaking at 2-4 hr p.i. (similar as STm) (Figure 4C). In contrast, Ab infection did not significantly increase surface and intracellular CCR10 expression (Figure 4—figure supplement 1C,D). in vivo, ~30% of neutrophils from Ab-infected mice displayed surface CCR3 expression (Figure 4F) and very few expressed surface CCR10 (Figure 4—figure supplement 1F).

Additionally, we tried exposing bone marrow neutrophils to CCL28 (50nM) and co-stimulating with CCL28 and STm (MOI=10) for 1 hour; however, we did not find a significant increase of CCR3 and CCR10 surface expression with CCL28 exposure (see Author response image 2). These results track with the finding that CCR3 expression levels of mucosal neutrophils during infection were independent of CCL28 status.

**Author response image 2. sa2fig2:** CCL28 exposure does not alter surface expression of CCR3 or CCR10 by neutrophils. Enriched bone marrow neutrophils (1x10^6^ cells/well) were infected in vitro with NMSopsonized STm (MOI=10) or mock-infected with vehicle (DPBS+10%NMS), with or without coincubation with recombinant mouse CCL28 (50 nM) for 1 hr. Cells were then stained and profiled by flow cytometry for CCR3 and CCR10 expression by live neutrophils. Connected symbols represent data from bone marrow cells extracted from the same mouse +/- CCL28 exposure, bars represent the mean.